# Impact of carbonate saturation on large Caribbean benthic foraminifera assemblages

Ana Martinez[1], Laura Hernández-Terrones[2], Mario Rebolledo-Vieyra[3], Adina Paytan[4]

[1] Department of Earth and Planetary Sciences, University of California Santa Cruz, 1156 High Street, Santa Cruz, CA 95064, USA

[2] Universidad del Caribe, L-1. Mz 1, Esq. Fracc. Tabachines SM 78, Cancún, Quintana Roo, 77528, México

[3] Chipre 5, Resid. Isla Azul, Cancún, Quintana Roo, 77500, México

[4] Institute of Marine Science, University of California Santa Cruz, Santa Cruz, CA 95064, USA

*Correspondence to*: Adina Paytan (apaytan@ucsc.edu)

**Abstract.** Increasing atmospheric carbon dioxide and its dissolution in seawater have reduced ocean pH and carbonate ion concentration with potential implications to calcifying organisms. To assess the response of large Caribbean benthic foraminifera to low carbonate saturation conditions, we analysed benthic foraminifers' abundance and relative distribution in surface sediments in proximity to low carbonate saturation submarine springs and at adjacent control sites. Our results show that total abundance of large benthic foraminifera was significantly lower at the low pH submarine springs than at control sites, although responses were species-specific. The relative abundance of high magnesium, porcelaneous foraminifera was higher than that of hyaline foraminifera at the low pH springs due to the abundant *Archaias angulatus,* a chlorophyte-bearing foraminifer which secretes a large and robust test that is more resilient to dissolution to low calcite saturation. The different assemblages found at the submarine springs indicate that calcareous symbiont-barren foraminifera are more sensitive to the effects of ocean acidification than agglutinated and symbiont-bearing foraminifera, suggesting that future ocean acidification will likely impact natural benthic foraminifera populations.

## 1 Introduction

Anthropogenic activities such as deforestation and fossil fuel burning are increasing the concentration of carbon dioxide ($CO_2$) in the atmosphere. About one third of all the $CO_2$ emitted into the atmosphere by humans over the past 200 years has been absorbed by the oceans (Sabine et al., 2004) causing a change in ocean chemistry, lowering the pH and the concentration of carbonate ions in seawater, collectively referred to as ocean acidification. It is expected that ocean pH will decrease even more, by ~0.4 pH units by year 2100 (Caldeira and Wickett, 2003; Orr et al., 2005) with possible consequences to marine organisms and ecosystems (Raven et al., 2005). Marine calcifying organisms may be particularly sensitive due to the lower availability of carbonate ions which are required for their shell formation (Raven et al., 2005).

Foraminifera are single celled organisms that are abundant in the marine water column and sediments, playing key roles in many marine ecosystems including being basal contributors to the marine food web and essential elements of the marine carbonate pump (Legendre and Le Fèvre, 1995; Culver and Lipps, 2003; Hain et al., 2014). Calcareous foraminifera produce calcium carbonate tests of diverse shapes and thickness while agglutinated foraminifera build a test made of detrital particles

and thecate foraminifera lack a test. The calcification pathway and magnesium content of calcareous foraminifera varies between perforate hyaline and imperforate porcelaneous foraminifera (Brasier, 1980). Some large benthic foraminifera harbour photosynthetic algal symbionts while others rely solely on heterotrophic feeding (Murray, 1991). The diversity of life styles and test characteristics suggest that the sensitivity of this group of organisms to changing ocean carbonate chemistry will be species dependent (Fabry et al., 2008; Fujita et al., 2011).

Laboratory culture experiments where benthic foraminifera were maintained under controlled conditions (i.e. partial pressure of $CO_2$, alkalinity, etc.) generally showed a decline in foraminifera calcification under high $pCO_2$ (Erez, 2003; Haynert et al., 2011; Keul et al., 2013). However, this response was not uniform and varied among species (Fujita et al., 2011; Hikami et al., 2011; McIntyre-Wressnig et al., 2013). Field studies at $CO_2$ vents in the Pacific Ocean (Fabricius et al., 2011; Uthicke et al., 2013) and Mediterranean Sea (Dias et al., 2010) reported a decrease in benthic foraminiferal abundance with increasing $pCO_2$,

especially of calcareous species; nonetheless benthic foraminifera have been found living near $CO_2$ vents in the northern Gulf of California (Pettit et al., 2013) and near experimentally injected deep-sea $CO_2$ hydrate (Bernhard et al., 2009) and generally foraminifera can be found in a wide range of environments (Brasier, 1980).

To shed light on the potential response of large Caribbean benthic foraminifera to future increase of $CO_2$ concentration and associated decrease in pH and carbonate ion concentrations, we studied the absolute and relative abundance of large benthic

foraminifera living around a series of submarine springs that naturally discharge low carbonate saturation state ($\Omega$) saline groundwater in the Yucatan Peninsula (Mexico) (Crook et al., 2012). The Yucatan peninsula is a karstic region with extensive nearshore submarine groundwater springs that discharge water characterized by low pH and high total inorganic carbon and total alkalinity, but only slightly lower salinity and similar temperatures to local marine conditions (Hofmann et al., 2011; Crook et al., 2012; Crook et al., 2013; Paytan et al., 2014; Null et al., 2014; Crook et al., 2016). Previous studies have

determined that the springs have been discharging low $\Omega$ water for millennia (Back et al., 1979); therefore, they serve as a natural laboratory to study the *in-situ* responses of marine organisms and ecosystems to long-term exposure to low $\Omega$ which may not be captured in short-term experiments (Andersson et al., 2015). Field studies from this site reported reduced coral species richness and coral colony size at the springs compared to control sites (Crook et al., 2012) and 70% less cover of calcifying benthic organisms after 14 months of recruitment experiment (Crook et al., 2016). We hypothesize that benthic

foraminifera assemblages will also differ between the springs and control sites, decreasing in overall abundance and having distinct species composition depending on test type, magnesium content, feeding strategy and photosymbiotic associations of foraminifera.

## 2 Materials and methods

### 2.1 Field sampling

Benthic foraminifera from the upper centimetre of sediment were collected with a spoon and stored in centrifuge tubes in October 2011 near five submarine groundwater springs (Norte, Mini, Pargos, Laja and Gorgos) at Puerto Morelos reef Lagoon (National Marine Park), in the Mexican Caribbean coast off Quintana Roo (Fig. 1). At each spring site, five replicates of surface sediment samples (coarse sand) were collected along with water samples, from near the centre of the submarine spring and at five control sites about two meters away from each spring, outside the impact area of the spring.

### 2.2 Water chemistry

Water temperature and pH were measured in situ with a handheld YSI analyzer (Yellow-spring model 63). Seawater samples were filtered (0.2 μm filter) and split into aliquots for total inorganic carbon ($C_T$), total alkalinity ($A_T$) and salinity measurements following the standard operating procedures described by (Dickson et al., 2007). Total inorganic carbon was analyzed on a CM5011 Carbon Coulometer (UIC, Inc.; analytical measurement error: $\pm$ 3 μmol kg$^{-1}$). Total alkalinity was measured using an automated open-cell, potentiometric titrator (Orion model 950; analytical measurement error: $\pm$ 2 μmol kg$^{-1}$). Certified $CO_2$ reference material (from A. Dickson lab at UC San Diego, batch 112) was used to calibrate the instruments. Salinity was analyzed using a portable salinometer (Portasal Model 8410, Guild Line). The program $CO_2$Sys (Pierrot et al., 2006) was used to calculate pH, carbonate ion concentrations and the $\Omega$ of seawater ($CO_2$ dissociation constants: (Lueker et al., 2000); KHSO$_4$: Dickson; B concentration: (Uppström, 1974)).

### 2.3 Foraminiferal analysis

Five replicate sediment samples per site were freeze dried, weighed, washed with deionized water through a 63 μm sieve to remove clay and silt, dried at 50°C and the >250 μm fraction analyzed under an optical microscope (Bausch and Lomb) to determine foraminiferal abundance measured as individuals per gram of sediment. The >250 μm fraction contains the assemblage of adult individuals which are likely to be conserved in the sediment (Martin, 1986). Small juveniles of species dominating shallow coastal setting have high mortality rates (pre-productive death rate of 99.5% for *A. angulatus*, (Knorr et al., 2015); >99% for Amphistegina spp., (Muller, 1974) and mortality rates of large foraminifera drop once their diameter is ~ 0.5 mm (Hallock and Glenn, 1986). Specifically in our samples the >250 μm fraction typically constituted >80% of the total tests in a sample. Indeed, large-size foraminifera are typical for warm, oligotrophic, well-lit, shallow water assemblages (Hallock, 1985). At least 1 gram of sediment per replicate was analyzed (with 2 grams per replicate for most samples). At least 300 individuals per replicate were picked; however, in 24 of the 50 samples less than 300 individuals per replicate were picked due to low foraminifera abundance. Foraminifera were identified following several taxonomic references (d'Orbigny, 1839; Poag, 1981; Wantland, 1967; Crevison and Hallock, 2001), each individual within a genus was counted, and total foraminiferal

and genus abundances were normalized to sediment weight. Only the most common genera (>5% of the assemblage in 10% of the samples) were picked and considered for statistical analyses.

### 2.4 Test weight

Tests of *Discorbis rosea* from the 250-355 μm sediment size fraction (2 to 122 individuals) were weighted using an analytical micro-balance (Sartorius, model CP2P, ± 5 μg error) and average weight per specimen determined. This species was chosen because of its abundance in most of the samples and the relatively constant test size.

### 2.5 Statistical analysis

Data analysis and visualization were performed using R program version 3.4.3 (Team 2017) and "vegan" package in R (Oksanen et al., 2013). Non-parametric Mann-Whitney rank sum test was conducted to determine differences in foraminiferal abundance and weight between each low $\Omega$ submarine spring and its corresponding control site. Permutational multivariate analysis of variance (PERMANOVA, 9,999 permutations) was used on the Bray-Curtis dissimilarity matrix after the square root transformed relative abundance of foraminifera to test for differences in community structure between saturation states and sites. Similarity percentages analysis (SIMPER) was used to determine the most important genera that contributed to dissimilarities in community structure. Nonmetric multidimensional scaling (nMDS) ordination was used to visualize the similarity in foraminiferal assemblages among $\Omega$ levels and sites. nMDS plots were created with metaMDS function on Bray-Curtis dissimilarity matrix of foraminiferal relative abundances and constrained to 2 dimensions. To evaluate the effects of environmental variables on foraminiferal relative abundance, the log-transformed water chemistry data was overlaid using envfit function of vegan library (Dixon, 2003) with 999 permutations.

### 3 Results

### 3.1 Water chemistry

The $\Omega$, pH and salinity of water in all springs was lower than their corresponding control sites (Table 1), while alkalinity ($A_T$) and total inorganic carbon ($C_T$) were higher than control sites. Temperature (T) was similar at all locations. These data represent the analyses of discrete water samples collected during sediment sampling; more data including continuous data collected by deployed sensors at some of these sites have been previously published (Crook et al., 2012; Crook et al., 2013; Crook et al., 2016; Null et al., 2014; Paytan et al., 2014; Hofmann et al., 2011) and data reported here are within the range of the published data. The specific spring sites were selected because the salinity at these sites is >30 over 90% of the time and it does not drop below 27; when salinity drops below 30 psu (7% of the time), the low salinity exposure lasts for very short periods of time always less than 1 hour (Crook et al., 2013)

## 3.2 Absolute abundance of foraminifera

Absolute abundance of foraminifera measured as total number of individuals per gram of sediment was higher at high $\Omega$ control sites than at low $\Omega$ springs in Norte (W = 25, p < 0.01), Mini (W = 25, p < 0.01), Pargos (W = 25, p < 0.01) and Laja (W = 25, p < 0.01) but not in Gorgos (W = 21, p = 0.095) (Fig. 2).

## 3.3 Genus assemblage

The seven most abundant genera were: *Amphistegina*, *Archaias*, *Asterigerina*, *Quinqueloculina*, *Triloculina*, *Discorbis* and *Gaudryina*. Other foraminifera that were present in some of the samples at a smaller abundance (<5% of assemblage) belong to the following genera: *Borelis*, *Clavulina*, *Elphidium*, *Spiroloculina*, *Peneroplis*, *Laevipeneroplis*, *Planorbulina, Sorites*, *Vertebralina* and *Heterostegina*. The composition of foraminifera communities (relative abundance of genera) changed significantly between saturation states (PERMANOVA$_{saturation}$, $F_{1,50} = 12.11$, p < 0.0001) and between sites (PERMANOVA$_{site}$, $F_{4,50} = 8.15$, p < 0.0001). SIMPER analysis revealed that *Archaias* and *Discorbis* genera contributed the most to dissimilarities in community structure between low $\Omega$ and high $\Omega$ in most of the sites while *Asterigerina* contributed the most in Pargos (Fig. 3). *Archaias* relative abundance increased at low $\Omega$ and *Discorbis* and *Asterigerina* relative abundances decreased at low $\Omega$ in all sites. *Amphistegina* and *Gaudryina* relative abundances increased at low $\Omega$ in all sites but Norte. *Quinqueloculina* and *Triloculina* combined relative abundance decreased at low $\Omega$ in Pargos, Laja and Gorgos and increased in Norte and Mini.

## 3.4 Foraminifera test type

Foraminifera were divided into three groups to investigate abundance differences based on test type. The calcareous porcelaneous group included *Archaias angulatus* and several species of *Quinqueloculina* and *Triloculina* genera. The calcareous hyaline group included *Amphistegina*, *Asterigerina* and *Discorbis*. The non-calcareous agglutinated group included individuals of the genus *Gaudryina*. Porcelaneous absolute abundance was lower at low $\Omega$ at all sites but Gorgos (Fig. 4) (Norte: W = 23, p < 0.05; Mini: W = 25, p < 0.01; Pargos: W = 25, p < 0.01; Laja: W = 25, p < 0.01; Gorgos: W = 20, p = 0.151). Hyaline absolute abundance was lower at low $\Omega$ at all sites (Norte: W = 25, p < 0.01; Mini: W = 25, p < 0.01; Pargos: W = 25, p < 0.01; Laja: W = 25, p < 0.01; Gorgos: W = 25, p < 0.01). The absolute abundance of agglutinated foraminifera was lower at low $\Omega$ than at high $\Omega$ in Norte (W = 24, p < 0.05) and Mini (W = 25, p < 0.01) and did not vary with $\Omega$ in Pargos (W = 16, p = 0.548), Laja (W = 21, p = 0.095), and Gorgos (W = 11, p = 0.841).

Relative abundance of foraminifera measured as a percentage of each group within the population also differed between $\Omega$ conditions (Fig. 4). Porcelaneous relative abundance was higher at low $\Omega$ in Norte and Laja (Norte: W = 0, p < 0.01; Mini: W = 5, p = 0.151; Pargos: W = 5, p = 0.151; Laja: W = 0, p < 0.01; Gorgos: W = 5, p = 0.142). In contrast, the hyaline relative abundance was lower at low $\Omega$ in Norte and Laja (Norte: W = 25, p < 0.01; Mini: W = 20, p = 0.142; Pargos: W = 20, p = 0.151; Laja: W = 25, p < 0.01; Gorgos: W = 20, p = 0.151). The relative abundance of agglutinated foraminifera was higher at

low $\Omega$ in Laja (W = 2, p = 0.05) and did not vary with $\Omega$ in the other four sites (Norte: W = 16, p = 0.548; Mini: W = 6, p = 0.222; Pargos: W = 3, p = 0.056; Gorgos: W = 7, p = 0.310).

### 3.5 Magnesium content in test of calcareous foraminifera

Calcareous foraminifera were divided into three groups based on their magnesium content of their test to evaluate the effect of $\Omega$ state on abundance. Foraminifera were grouped into low (*Discorbis*), intermediate (*Amphistegina* and *Asterigerina*) and high Mg content (*Archaias*, *Quinqueloculina* and *Triloculina*) tests. The absolute abundance of foraminifera with low Mg test was lower at low $\Omega$ in all sites (Fig. 5) (Norte: W = 25, p < 0.01; Mini: W = 25, p < 0.01; Pargos: W = 25, p < 0.01; Laja: W = 25, p < 0.01; Gorgos: W = 25, p < 0.01). Similarly, the absolute abundance of intermediate Mg foraminifera was lower at low $\Omega$ in all sites (Norte: W = 25, p < 0.01; Mini: W = 25, p < 0.01; Pargos: W = 25, p < 0.01; Laja: W = 25, p < 0.01; Gorgos: W = 23, p < 0.05). The absolute abundance of high Mg foraminifera was lower at low $\Omega$ at all sites but Gorgos (Norte: W = 23, p < 0.05; Mini: W = 25, p < 0.01; Pargos: W = 25, p < 0.01; Laja: W = 25, p < 0.01; Gorgos: W = 20, p = 0.151).

The relative abundance of low Mg foraminifera was lower at low $\Omega$ in Norte, Mini and Laja (Norte: W = 25, p < 0.01; Mini: W = 25, p < 0.01; Pargos: W = 20, p = 0.151; Laja: W = 25, p < 0.01; Gorgos: W = 20, p = 0.151). The relative abundance of intermediate Mg foraminifera was significantly lower at low $\Omega$ in Norte and Pargos (Norte: W = 25, p < 0.01; Mini: W = 8, p = 0.421; Pargos: W = 23, p < 0.05; Laja: W = 18, p = 0.309; Gorgos: W = 20, p = 0.151). In contrast, the relative abundance of high Mg foraminifera was higher at low $\Omega$ in Norte and Laja (Norte: W = 0, p < 0.01; Mini: W = 5, p < 0.142; Pargos: W = 5, p < 0.151; Laja: W = 0, p < 0.01; Gorgos: W = 5, p = 0.151).

### 3.6 Feeding strategy of calcareous foraminifera

Calcareous foraminifera were divided into two groups based on their feeding strategy: heterotrophic, symbiont-barren foraminifera and symbiont-bearing foraminifera. The absolute abundance of calcareous heterotrophic foraminifera was lower at low $\Omega$ than at high $\Omega$ at all sites but Gorgos (Fig. 6) (Norte: W = 25, p < 0.01; Mini: W = 25, p < 0.01; Pargos: W < 0.05; Laja: W = 25, p < 0.01; Gorgos: W = 20, p = 0.151). The absolute abundance of symbiont-bearing foraminifera was also lower at low $\Omega$ than at high $\Omega$ at all sites but Gorgos (Norte: W = 24, p < 0.05; Mini: W = 25, p < 0.01; Pargos: W = 25, p < 0.01; Laja: W < 25, p < 0.01; Gorgos: W = 19, p = 0.222). The relative abundance of heterotrophic foraminifera was lower at low $\Omega$ than at high $\Omega$ in all sites but Gorgos (Norte: W = 25, p < 0.01; Mini: W = 25, p < 0.01; Pargos: W = 25, p < 0.01; Laja: W = 25, p < 0.01; Gorgos: W = 20, p = 0.151). In contrast, the relative abundance of symbiont-bearing foraminifera was higher at low $\Omega$ at all sites but Gorgos (Norte: W = 0, p < 0.01; Mini: W = 0, p < 0.01; Pargos: W = 0, p < 0.01; Laja: W = 0, p < 0.01; Gorgos: W = 5, p = 0.151).

### 3.7 Symbiont type of calcareous foraminifera

To test the differences among symbiont types on foraminifera abundance at low $\Omega$, symbiont-bearing foraminifera were divided into two groups: diatom-bearing foraminifera (*Amphistegina* and *Asterigerina*) and chlorophyte-bearing foraminifera (*Archaias*). The absolute abundance of diatom-bearing foraminifera was lower at low $\Omega$ at all sites (Fig. 7) (Norte: W = 25, p < 0.01; Mini: W = 25, p < 0.01; Pargos: W = 25, p < 0.01; Laja: W = 25, p < 0.01; Gorgos: W = 23, p < 0.05). The absolute abundance of chlorophyte-bearing foraminifera was lower at low $\Omega$ in Mini, Pargos and Laja and did not vary significantly in Norte and Gorgos (Norte: W = 20, p = 0.151; Mini: W = 25, p < 0.01; Pargos: W = 24, p < 0.05; Laja: W = 25, p < 0.01; Gorgos: W = 12, p = 1).

The relative abundance of diatom-bearing foraminifera was lower at all sites but Mini (Norte: W = 25, p < 0.01; Mini: W = 17, p = 0.421; Pargos: W = 24, p < 0.05; Laja: W = 25, p < 0.01; Gorgos: W = 25, p < 0.01). Contrastingly, the relative abundance of chlorophyte-bearing foraminifera was higher at all sites but Mini (Norte: W = 0, p < 0.01; Mini: W = 8, p < 0.421; Pargos: W = 1, p < 0.05; Laja: W = 0, p < 0.01; Gorgos: W = 0, p < 0.01).

### 3.8 Environmental factors

The nMDS plots showed a clear clustering of relative abundances between high and low $\Omega$, while this clustering was not apparent between sites at a specific saturation state (Fig. 8). The envfit function revealed that areas where calcareous heterotrophic foraminifera were relatively more abundant are the control sites, which are characterized by higher pH ($R^2$ = 0.3531, p = 0.001), salinity ($R^2$ = 0.4420, p = 0.001), and $\Omega$ (represented as the arrow titled calcite in Fig. 8, $R^2$ = 0.4735, p = 0.001), while areas where calcareous heterotrophic foraminifera were less abundant are the spring sites which are characterized by higher alkalinity (represented as arrow A in Fig. 8, $R^2$ = 0.4420, p = 0.001), and higher total inorganic carbon (represented as arrow C in Fig. 8, $R^2$ = 0.4261, p = 0.001). Calcareous symbiont-bearing foraminifera were relatively more abundant in low $\Omega$ areas (blue symbols) with higher temperature (represented as arrow T in Fig. 8, $R^2$ = 0.1234, p = 0.036), although the temperature is not on the main gradient of variation and the difference among sites was at most two degrees Celsius, which is lower than diurnal or seasonal natural variability within sites. Relative abundance of agglutinated foraminifera did not seem to be affected by the main gradient explaining the maximal variance of data. These trends are consistent with field observations.

### 3.9 Test weight

The average test weights of *Discorbis rosea* (size fraction 250-355 µm) did not differ among saturation states in any of the sites (Norte: W = 13, p = 1; Mini: W = 13, p = 0.2; Pargos: W = 7, p = 0.309; Laja: W = 8, p = 0.421; Gorgos: W = 20, p = 0.151) (Fig. 9).

## 4 Discussion

### 4.1 Absolute abundance of calcifying benthic foraminifera decreases at low $\Omega$ springs

The analysis of foraminiferal abundance in surface sediments collected from low $\Omega$ submarine springs and control sites revealed that the absolute abundance of calcareous foraminifera was lower at springs than at control sites (Fig. 2). Calcification

of calcareous foraminifera is a process that depends on the carbonate chemistry of seawater and requires calcite supersaturated conditions at the calcification site (Erez, 2003; Bentov et al., 2009). Foraminifera endocytose seawater to bring calcium and inorganic carbon to the active calcification site (Bentov et al., 2009). In the process, the vacuolized seawater is alkalinized to a pH of ~9 to overcome magnesium mediated inhibition of calcite precipitation and to promote the conversion of inorganic carbon speciation from bicarbonate to carbonate ions (de Nooijer et al., 2009). This pH elevation at the site of calcification is

achieved by using ATP to pump protons out of the foraminifera protoplasm (Glas et al., 2012b; Toyofuku et al., 2017). If the ambient pH is low, the foraminifera have to devote more energy to rising the intracellular pH to promote calcification, making the conditions at low pH sites less favorable for calcification (de Nooijer et al., 2009). Indeed, this may explain the decrease we see in the total abundance of calcareous porcelaneous and hyaline foraminifera at the low pH, low $\Omega$ submarine springs. Agglutinated foraminifera absolute abundance was similar between springs and control sites in three of the five sampled sites,

and their relative abundance was similar among springs and controls in four of the five sites (Fig. 4), although their abundance was overall low in both springs and control sites. Furthermore, SIMPER analysis revealed that agglutinated *Gaudryina* foraminifera relative abundance increased at low $\Omega$ in most of the sites (Fig. 3). Since agglutinated foraminifera tests are not made of calcium carbonate, they may be less influenced by the low $\Omega$ seawater at the springs than calcareous foraminifera. A lesser impact of low pH on agglutinated foraminifera abundance has also been observed in foraminifera present at $CO_2$ vents

at Papua New Guinea (Uthicke et al., 2013) and Ischia, in the Mediterranean Sea (Dias et al., 2010). Similarly, the abundance of non-calcifying thecate and agglutinated foraminifera living in direct contact with experimentally injected $CO_2$ hydrate did not decline significantly with decreasing pH (Bernhard et al., 2009). However, species-specific survival rates of agglutinated foraminifera during a laboratory experiment at 2000 ppm of $p$CO$_2$ suggests that other agglutinated species different than *Gaudryina* may react in a different manner to low $\Omega$ (van Dijk et al., 2017).

Since many environmental parameters co-vary in natural environments (Andersson et al., 2015), including at our field site, it is possible that the trends in absolute and relative abundances of foraminifera present at the springs are due to species-specific salinity preferences (the only other variable that consistently different at springs and control sites). The salinity of the discharging water at the sampled springs is > 30 for 93% of the time and it is constantly higher than 27 (Crook et al., 2013) as previously mentioned. Although the salinity tolerance ranges are not known for all the species found in the study area, many

foraminifera that are abundant in shallow warm coastal waters such as those at our sites, have a very wide salinity tolerance (Brasier, 1980). *Quinqueloculina* spp. has been found at salinity ranges of 12-35 with abundance peaks at 17 and 35 (Horton and Murray, 2007), *Amphistegina lessonii* has been kept between 25 and 45 in a lab experiment (Geerken et al., 2018) and *Archaias* has been reported to be present at salinities of 29-39 (Hallock and Peebles, 1993). Moreover, adaptation to changes

in salinity requires increased cellular osmoregulation (McLusky et al., 2004), which is expected to affect both agglutinated and calcareous foraminifera abundance. Since agglutinated foraminifera abundance is similar at the springs and control sites (Fig. 4) and does not seem to be affected by the main gradient of variation in carbonate chemistry and salinity (Fig. 8), we suggest that $\Omega$ and pH are the main drivers of calcareous foraminifera abundances seen in this study. Consistent with this conclusion,

the trends we see in absolute and relative abundance of calcareous and agglutinated foraminifera are in line with observations from other field studies where salinities did not differ between low and ambient pH sampling locations (Fabricius et al., 2011; Uthicke et al., 2013). Hence, the lower abundance of calcareous foraminifera we and others have observed in diverse settings with low $\Omega$ suggests that future reduction in $\Omega$ will negatively affect calcareous benthic foraminifera.

## 4.2 Porcelaneous, high-Magnesium tests foraminifera's relative abundance increases in low $\Omega$ springs

While absolute abundance of both porcelaneous and hyaline foraminifera was lower at low $\Omega$, a trend towards higher relative abundance of porcelaneous foraminifera and lower relative abundance of hyaline foraminifera is observed (Fig. 4). The higher relative abundance of porcelaneous (Fig. 4) and high magnesium foraminifera (Fig. 5) is driven by *Archaias angulatus*, which is the most common species found and contributes the most to community dissimilarity in all the sites (Fig. 3). *Archaias angulatus* is well preserved in sediments due to its robust, thick test (Hallock and Peebles, 1993) strengthened by crystal pillars

(Martin, 1986), and has been reported to account for more than 20% of the foraminiferal population in the South Florida shelf (Knorr et al., 2015), up to 54% of dead assemblages from North Florida Keys (Martin, 1986) and to be the most common species in Banco Chinchorro, South Yucatan Peninsula (Gischler and Möder, 2009). The lower relative abundance of hyaline, low magnesium foraminifera at low $\Omega$ (Fig. 4 and 5) is attributed to the decrease of *Discorbis* and *Asterigerina* (Fig. 3). These results are in contrast with the idea that porcelaneous, high magnesium foraminifera would be the "first responders" (Fujita et

al., 2011) to ocean acidification, since high Mg calcite is more soluble than low Mg calcite and aragonite at a given $pCO_2$ (Morse et al., 2006) and because Mg inhibits calcite crystallization. This can be attributed to the lower solubility of the robust tests.

The calcification pathway of perforate hyaline foraminifera (reviewed by de Nooijer et al., 2014) has been studied in more detail than the calcification process of porcelaneous foraminifera. Hyaline foraminifera capture ions through seawater

endocytosis (Bentov et al., 2009;de Nooijer et al., 2009) and transmembrane transport (Nehrke et al., 2013), and store them in separated intracellular reservoirs of inorganic carbon and calcium (Ter Kuile and Erez, 1991; Toyofuku et al., 2008). A perforated test is then secreted extracellularly within a primary organic sheet after intracellular Mg discrimination and pH increase of the vacuolized seawater to a pH of $\geq 9$ (Zeebe and Sanyal, 2002; Erez, 2003; de Nooijer et al., 2009). In contrast, porcelaneous foraminifera precipitate calcite needles inside intracellular vesicles (at a pH of ~ 9) and are later transported and

randomly assembled in an extracellular organic matrix to form a new test chamber (Angell, 1980; Hemleben et al., 1986; Erez, 2003; de Nooijer et al., 2009). These transporting vesicles have been reported to have a pH of 7.5-8.0 (de Nooijer et al., 2009). Since these vesicles have a lower pH, it is possible that less protons are pumped out of the vesicle. In addition, the lack of

internal calcium and inorganic carbon pools may require less energy to precipitate calcite tests, which can be a competitive advantage that explains the increase in relative abundance of porcelaneous foramnifera we see at low pH, low $\Omega$ springs. Another explanation, noted above, could be that lower dissolution rates of the more robust porcelaneous tests (Brasier, 1980; Schmiedl et al., 1997) results in the observed increase in the abundance of these tests. However, further research is needed to

test these hypotheses and to better understand the calcification pathway and preservation of porcelaneous foraminifera. These results can guide future controlled experiments in a laboratory setting.

### 4.3 Symbiont-bearing foraminifera increase in relative abundance at low $\Omega$ springs

The relative abundance of heterotrophic foraminifera decreased while the relative abundance of symbiont-bearing foraminifera increased in most of the springs (Fig. 6). Foraminifera hosting photosynthetic symbionts may be more resilient to low $\Omega$ since

they can access additional energy derived from photosynthates translocated from the algae (Hallock, 2000) to increase pH at the calcification site and for alkalinization of seawater vacuoles. In addition, symbiotic algae can promote calcification by removing foraminiferal metabolic N and P which impede crystal formation, by providing organic matter used to synthesize the organic matrix that precedes test growth (Fujita et al., 2011) and by increasing the pH on the surface of foraminifera (Glas et al., 2012a). These mechanisms may explain the significant increase in relative abundance of symbiont-bearing foraminifera

(>50% of the total calcareous population) while calcareous heterotrophic foraminifera relative abundance decreased (<50%) at low $\Omega$ springs. Although symbiont-bearing calcareous foraminifera were relatively more abundant than symbiont-barren foraminifera at low $\Omega$ sites, their absolute abundance decreased in comparison with sites at ambient $\Omega$, indicating that despite the symbionts, the conditions were less favorable than at ambient conditions. Short laboratory experiments with symbiont-bearing foraminifera cultured at high $p$CO$_2$ have reported reduced net calcification (Fujita et al., 2011; Hikami et al., 2011)

and tests dissolution signs (McIntyre-Wressnig et al., 2013). While photosynthetic activity may promote calcification, it does not fully compensate the deleterious effects of elevated $p$CO$_2$ on foraminifera calcification incubated in laboratory (Glas et al., 2012a) and field experiments (Uthicke and Fabricius, 2012). These studies suggest that benthic symbiont-bearing foraminifera can better survive at high $p$CO$_2$, but their calcification is reduced.

Foraminifera hosting chlorophytes (*Archaias*) were relatively more abundant at springs than those hosting diatoms

(*Amphistegina* and *Asterigerina*) (Fig. 7). Hyaline foraminifera hosting diatoms are thought to be more resilient to high $p$CO$_2$ than other symbiont-bearing foraminifers based on a meta-analysis of studies assessing the impacts of acidification on large benthic foraminifera (Doo et al., 2014). However, none of the studies included in the meta-analyses focused on chlorophyte-bearing foraminifera and due to the high variability in methodology, duration and species used in the experiments, it is not possible to make a direct comparison between these studies and an assemblage found at the natural low $\Omega$ springs in our study.

Foraminifera hosting chlorophytes may be more resilient to ocean acidification than those hosting diatoms, or the robustness of *Archaias* tests may be responsible for this difference in relative abundance. It is also plausible that the size of the symbiont-bearing foraminifera influences the survival and preservation under low $\Omega$ conditions. The relative abundance of *Asterigerina*

decreased at low Ω while *Amphistegina* increased, in spite of both being hyaline foraminifera hosting diatoms (Fig. 3). The larger size of *Amphistegina* in comparison to *Asterigerina* may allow for hosting a larger concentration of photosynthetic algae, as it has been suggested that the number of symbionts increases with test size (Hönisch and Hemming, 2004). In fact, *Archaias* has the largest tests of all the species found at the springs in this study. Furthermore, a larger size has been linked to reduced dissolution due to a smaller surface-volume ratio (Hönisch and Hemming, 2004), which may explain why large foraminifera overall are more abundant than smaller foraminifera at this location.

### 4.4 Discorbis rosea weight did not significantly vary among springs and control sites

The test weight of *D. rosea* did not significantly vary among springs and control sites. This lack of difference may be due to the large variability in test weight within populations and individuals. The variability in tests weights within a species may be due to differential individual growth rates (Fujita et al., 2011), body sizes (Henehan et al., 2017) or genotypes (Davis et al., 2017) with diverse calcification performance under the same Ω conditions. In our study, the weighted tests were all picked from the 250-355 μm sediment fraction and we took special care to select individuals of very similar size, however, each test was not normalized to shell diameter, hence the wide variability in test weights may be partially related to the range in test sizes.

### 4.5 Implications

The reduced absolute abundance of benthic foraminifera at low Ω springs suggest that there may be an overall decrease in benthic foraminifera abundance as a consequence of ocean acidification, with subsequent repercussions on the global carbon cycle and marine food web. *Archaias angulatus*, the most common species found in this study, is known to represent a large proportion of the foraminiferal population in different parts of the western tropical Atlantic Ocean (Martin, 1986; Gischler and Möder, 2009; Knorr et al., 2015), being the dominant large benthic foraminifera in the Florida-Bahamas carbonate province (Hallock et al., 1986). A laboratory study with *A. angulatus* reported a 50% decrease in growth rate after 28 days at pH 7.6, and an estimated reduction of 85% of carbonate production by this species in the South Florida reef tract and Florida bay, from 0.27 Mt/yr to 0.04 Mt/yr (Knorr et al., 2015). Besides changes in carbonate production, a decrease in foraminiferal abundance may have cascade effects through the ecosystem since foraminifera are an important link in the marine food web as they prey on bacteria and algae, and are predated on by many animals such as gastropods, bivalves, echinoderms and crustaceans (Culver and Lipps, 2003).

### 5 Conclusion

The absolute abundance of all large calcareous foraminifera decreased at springs discharging low Ω, low pH water. Porcelaneous, high magnesium foraminifera were relatively less impacted compared to hyaline foraminifera at the springs, possibly due to their different calcification mechanism and more robust tests and the lack of internal carbon and calcium pools.

The relative abundance of symbiont-bearing foraminifera increased while heterotrophic symbiont-barren foraminifera decreased under low $\Omega$ conditions, which may be explained by the higher energy availability provided by the symbiont to elevate the pH at the site of calcification. Chlorophyte-bearing foraminifera were relatively more abundant than diatom-bearing foraminifera. These trends are driven by the abundant large *Archaias angulatus*, a porcelaneous foraminifera hosting chlorophytes, which may be more resilient to low $\Omega$ due to its test robustness and large size that can lead to a higher concentration of symbiotic algae and reduced test dissolution. Further laboratory experiments are needed to confirm these results in a controlled setting without covarying environmental variables and to better understand the calcification pathway of porcelaneous foraminifera.

## 6 Author contribution

Conceived and designed the experiments: AM and AP. Conducted field work: AM AP, MRV, LH. Analyzed the data: AM and AP. Contributed reagents/materials/analysis tools: AM, AP, LH, MRV. AM and AP primarily wrote the paper and LH and MRV provided critical edits.

## 7 Competing interests

The authors declare that they have no conflict of interest

## 8 Acknowledgements

We are grateful to Daniel Mendoza, Patricia Domínguez, Caitlin Celic, Grace Moreno and Rob Franks for assistance during laboratory analysis and Yui Takeshita for assistance during field work. We would like to thank Joan Bernhard for insightful comments on this project and Ellen Thomas and Daniela Schmidt for advice regarding foraminifera identification and biology. The comments and suggestions of Inge van Dijk and two anonymous reviewers significantly improved this manuscript. This research was funded by the National Science Foundation-1040952 (to AP) and by the Ministry of Education of Spain Argo Global Grant and the Myers Oceanographic and Marine Biology Trust Grant (to AM). The funders had no role in study design, data collection and analysis, decision to publish, or preparation of the manuscript.

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

**10 Tables**

**Table 1: Carbonate chemistry parameters of discrete water samples collected at low saturation state submarine springs and adjacent high saturation state control sites (mean ± SD) at the time of sample collection ($A_T$= total alkalinity; $C_T$= total inorganic carbon).**

| Site | Depth (m) | | $A_T$ ($\mu mol\cdot kg^{-1}$) | $C_T$ ($\mu mol\cdot kg^{-1}$) | *pH | *$CO_3^{2-}$ ($\mu mol\cdot kg^{-1}$) | *$\Omega$ calcite | T (°C) | Salinity |
|---|---|---|---|---|---|---|---|---|---|
| Norte | 5.8 | control | 2354 ± 13 | 2051 ± 6 | 7.98 | 216.16 | 5.14 | 27.0 | 36.80 |
| | | spring | 2611 ± 3 | 2588 ± 3 | 7.38 | 67.03 | 1.66 | 27.5 | 32.21 |
| Mini | 4.9 | control | 2356 ± 3 | 2049 ± 6 | 7.99 | 218.13 | 5.16 | 26.4 | 37.3 |
| | | spring | 3108 ± 10 | 3197 ± 6 | 7.13 | 46.29 | 1.14 | 27.6 | 32.41 |
| Pargos | 6.8 | control | 2336 ± 4 | 2012 ± 12 | 8.01 | 229.56 | 5.49 | 27.6 | 36.17 |
| | | spring | 3000 ± 8 | 3048 ± 12 | 7.23 | 52.73 | 1.33 | 27.6 | 29.95 |
| Laja | 5.8 | control | 2357 ± 6 | 2092 ± 1 | 7.90 | 193.55 | 4.63 | 28.1 | 36.17 |
| | | spring | 2827 ± 9 | 2756 ± 10 | 7.51 | 102.65 | 2.50 | 27.9 | 32.75 |
| Gorgos | 7.2 | control | 2325 ± 3 | 2033 ± 3 | 7.96 | 209.44 | 5.02 | 27.8 | 35.90 |
| | | spring | 2874 ± 11 | 2987 ± 8 | 7.11 | 94.65 | 2.38 | 28.5 | 31.09 |

5   * Calculated using $CO_2Sys$

**11 Figures**

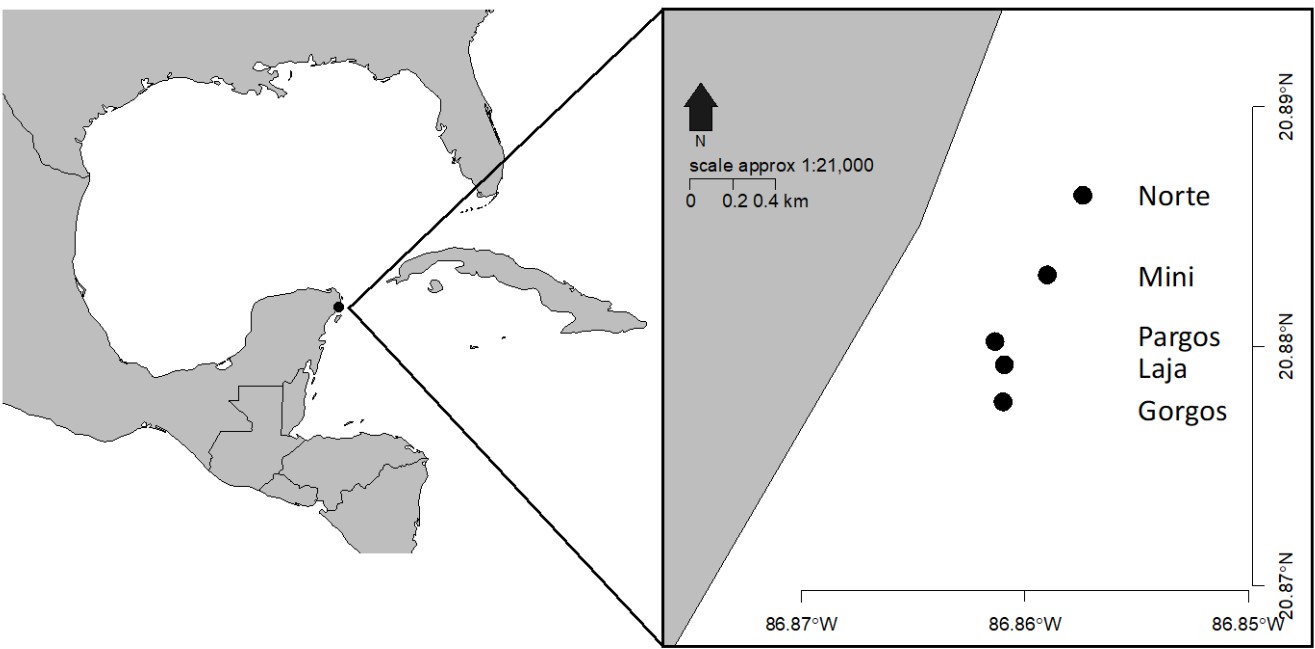

**Figure 1: Location of low carbonate saturation state submarine springs**

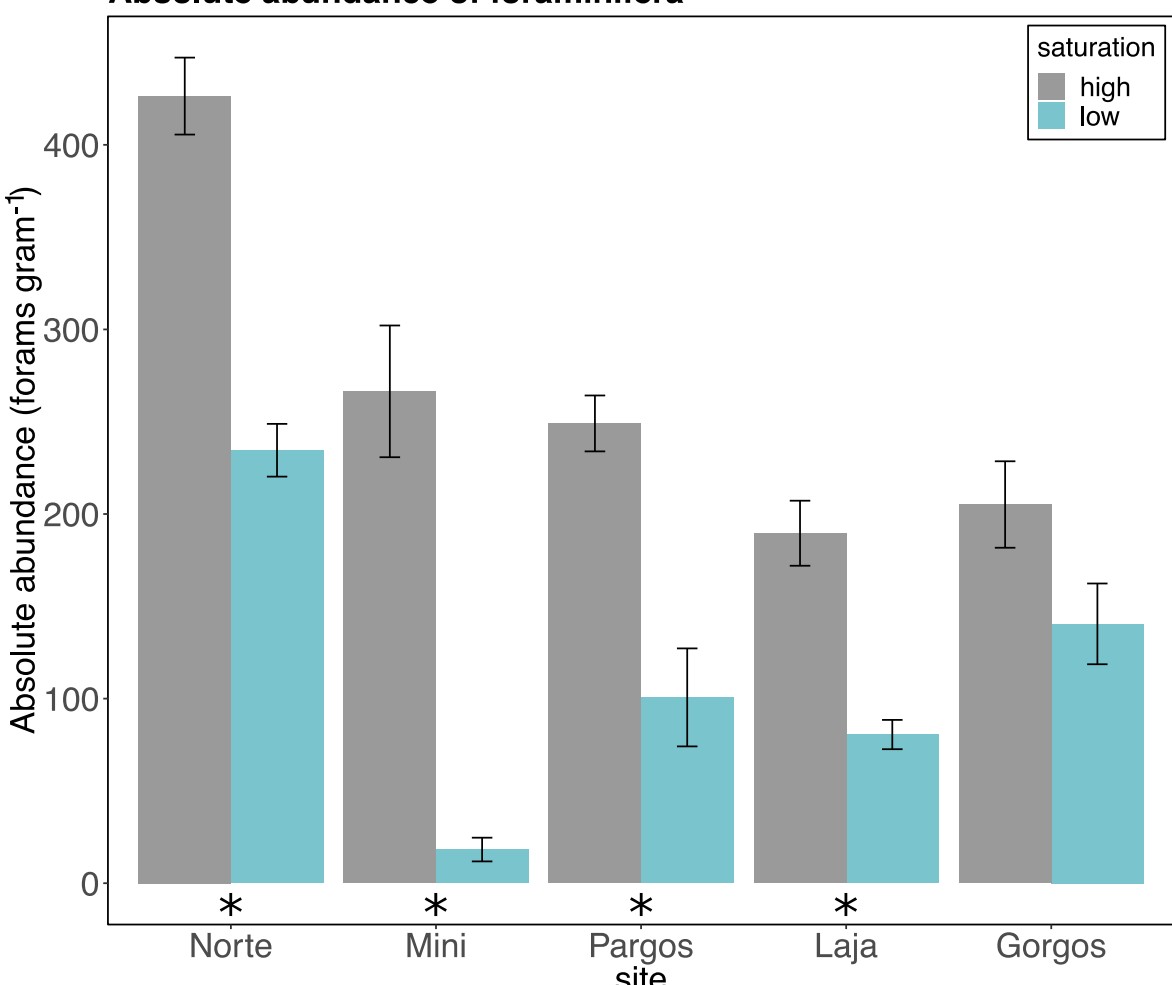

**Figure 2: Absolute abundance of foraminifera (number of specimens per gram of sediment) in different submarine springs (low saturation state) and their respective control sites (high saturation state). Data are mean ± SE (n= 5). The asterisk demarks a significant difference (p < 0.05) in abundance between paired springs and controls at each site according to Mann-Whitney rank sum test.**

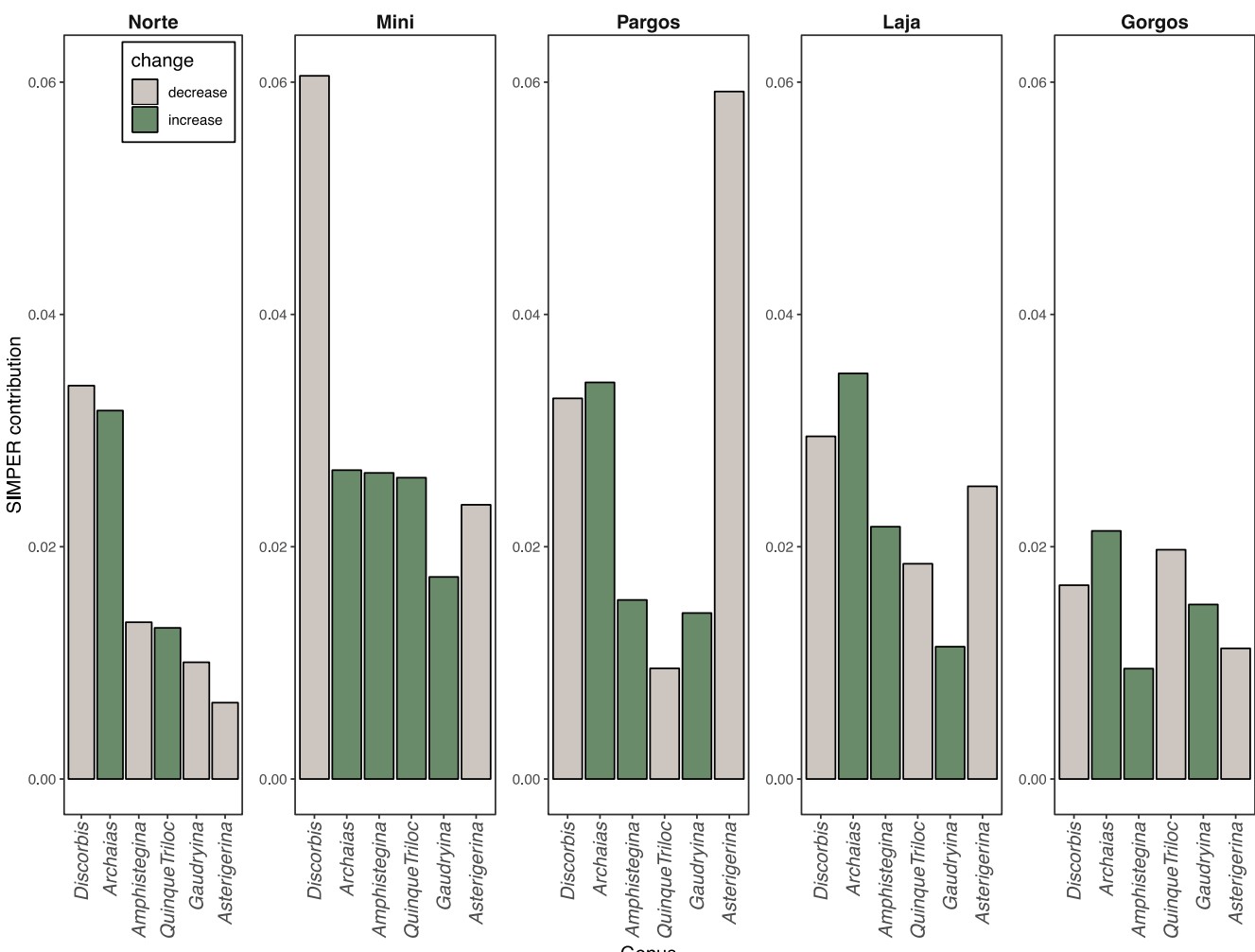

**Figure 3: SIMPER contribution of the most abundant genera. Bar height indicates the mean contribution of each genus to community dissimilarity. Green color represents an increase and grey color represents a decrease in the mean relative abundance of each genus at low saturation springs.**

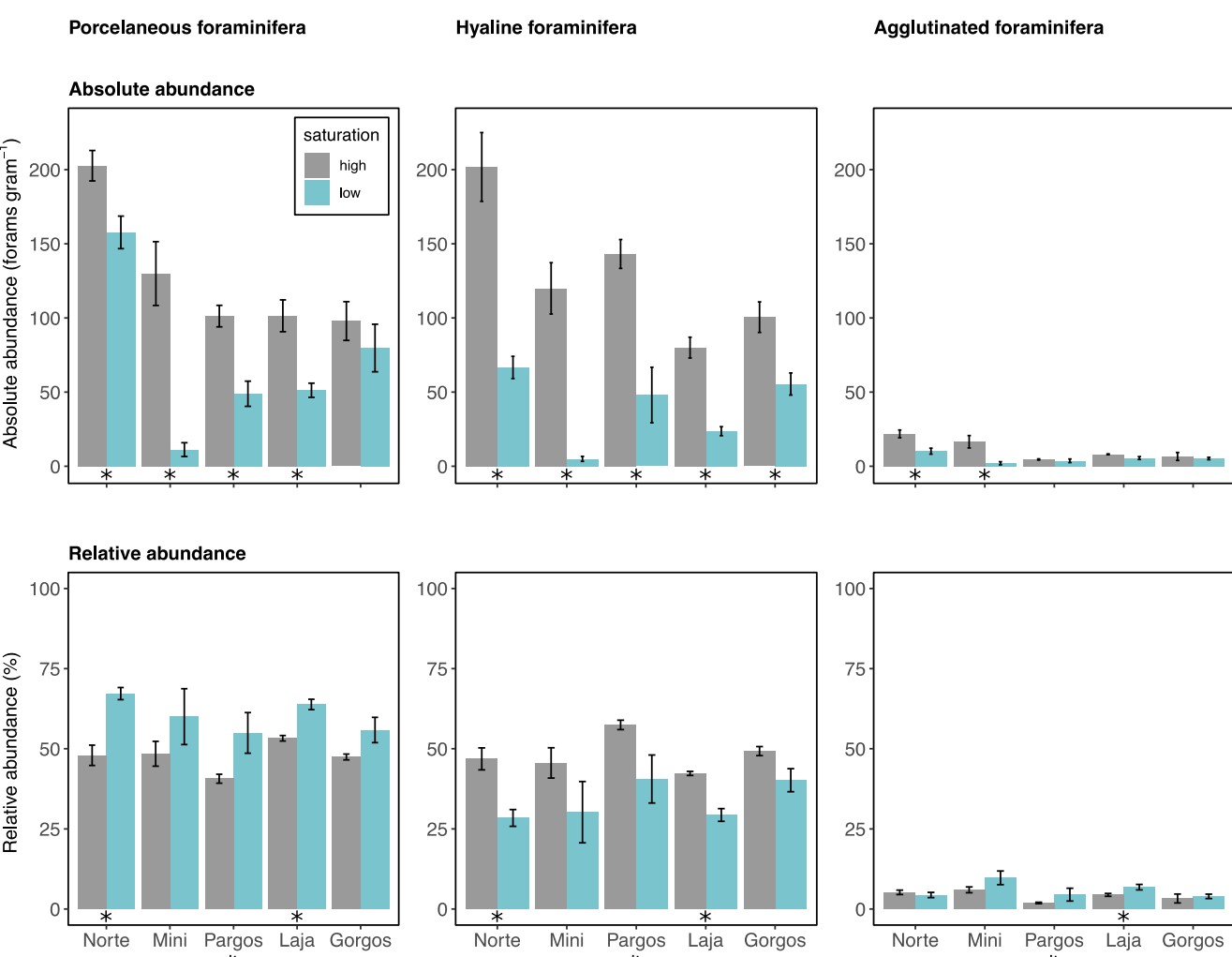

**Figure 4: Absolute abundance (specimens per gram of sediment) and relative abundance (percentage) of different foraminifera test types (porcelaneous, hyaline, and agglutinated). Data are mean ± SE (n= 5). The asterisk demarks a significant difference (p < 0.05) in abundance between paired springs and controls at each site according to Mann-Whitney rank sum test.**

**Figure 5: Absolute abundance (specimens per gram of sediment) and relative abundance (percentage) of foraminifera with different magnesium content tests. Data are mean ± SE (n= 5). The asterisk demarks a significant difference (p < 0.05) in abundance between paired springs and controls at each site according to Mann-Whitney rank sum test.**

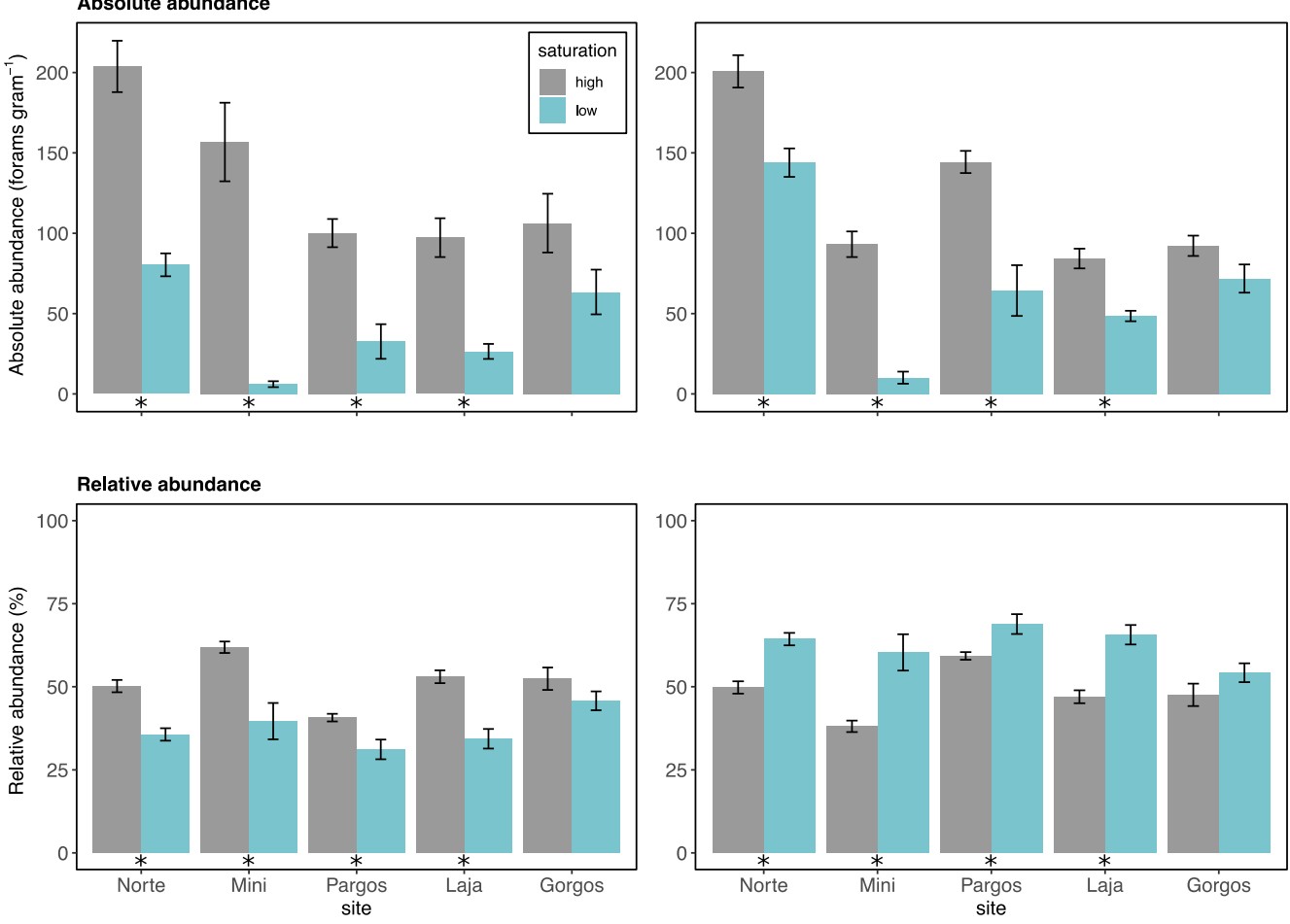

**Figure 6: Absolute abundance (specimens per gram of sediment) and relative abundance (percentage) of different feeding strategies of calcareous foraminifera (symbiont-barren heterotrophic and symbiont-bearing). Data are mean ± SE (n= 5). The asterisk demarks a significant difference (p < 0.05) in abundance between paired springs and controls at each site according to Mann-Whitney rank sum test.**

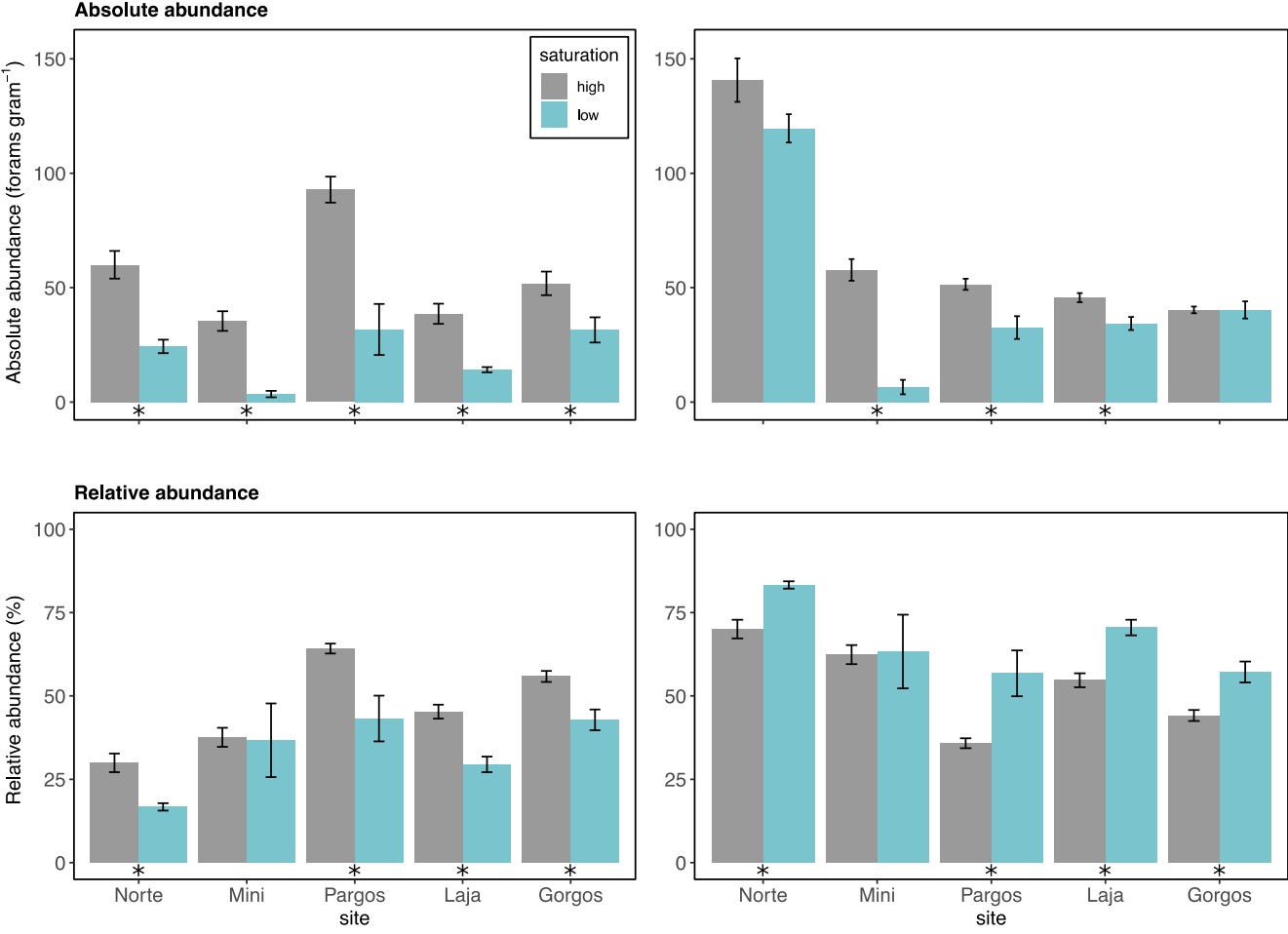

**Figure 7: Absolute abundance (specimens per gram of sediment) and relative abundance (percentage) of large calcareous foraminifera hosting different symbionts (diatoms and chlorophytes). Data are mean ± SE (n= 5). The asterisk demarks a significant difference (p < 0.05) in abundance between paired springs and controls at each site according to Mann-Whitney rank sum test.**

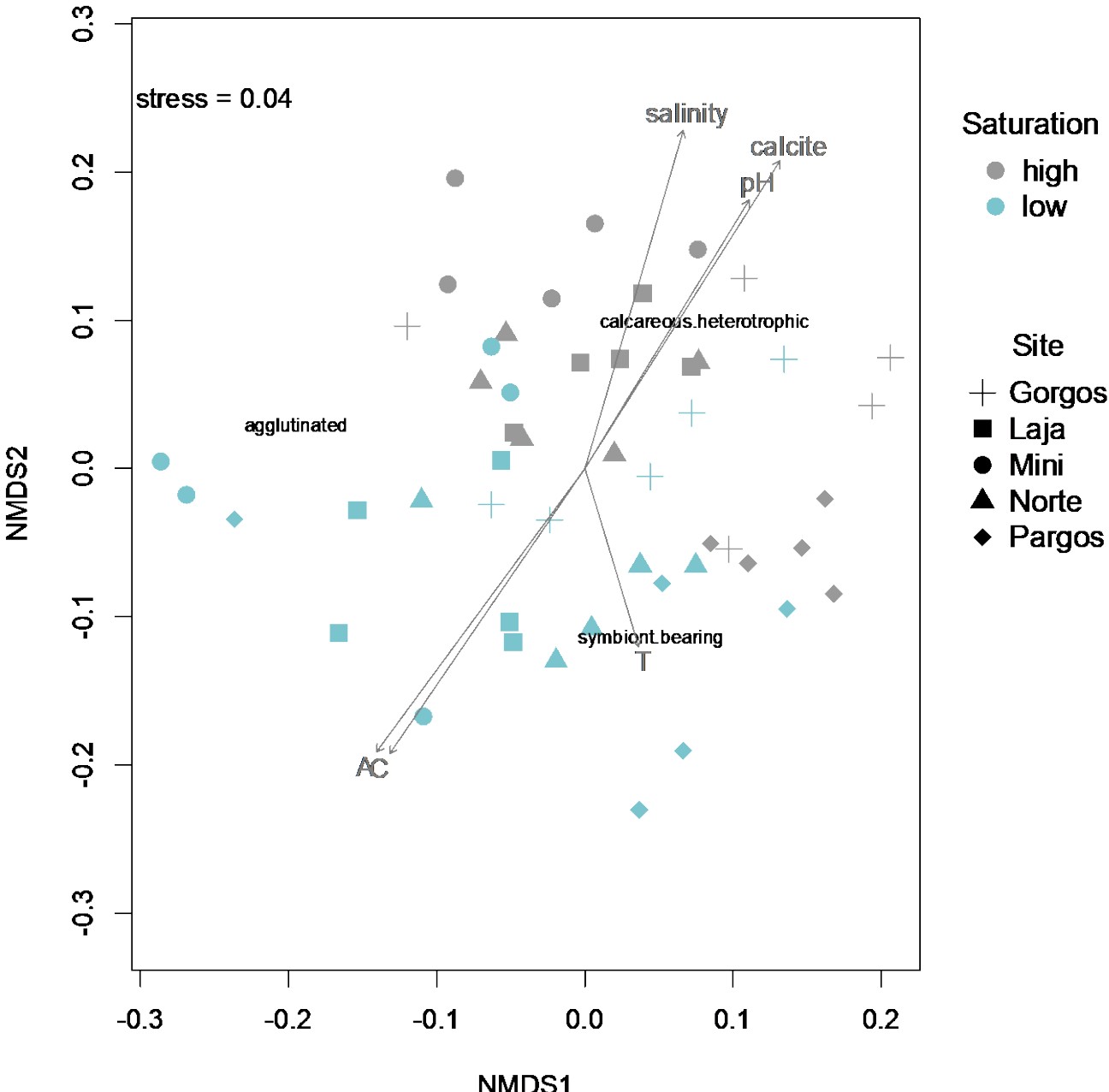

**Figure 8: Non-metric Multidimensional Scaling (nMDS) ordination plot for community structure (relative abundance) by carbonate saturation state and site with overlaid environmental parameters (A= total alkalinity; C= total inorganic carbon; T= temperature).**

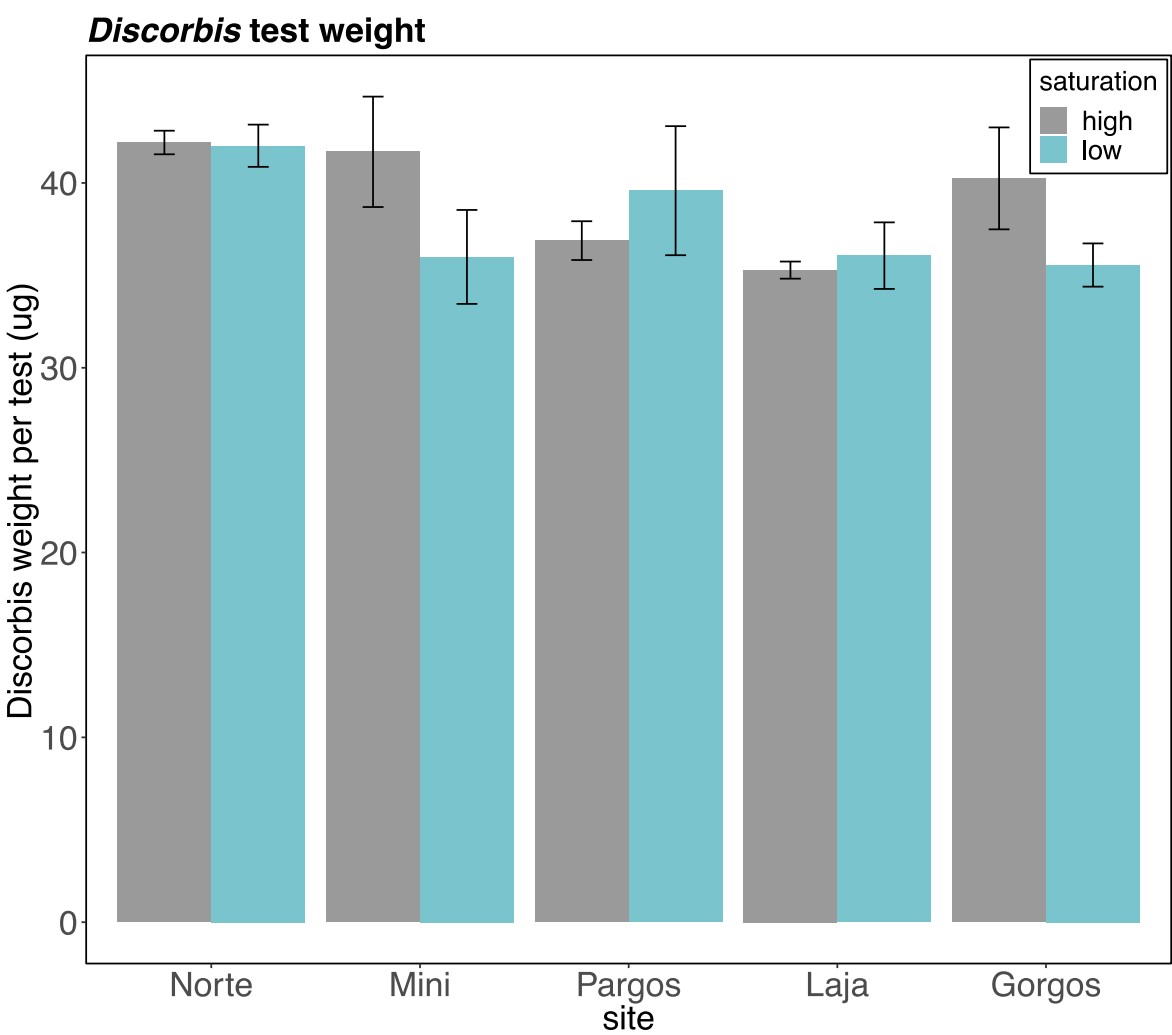

**Figure 9: Mean weight of *Discorbis rosea* tests (size fraction 250-355 μm) at low and high saturation at different submarine spring sites. Data are mean ± SE.**