# Peer review of "Impact of carbonate saturation on large Caribbean benthic foraminifera assemblages"

_Biogeosciences, 2018_

## Referee Comment (RC1) · I. van Dijk (Referee) · 7 Aug 2018

The manuscript 'Impact of carbonate saturation on large Caribbean benthic foraminifera assemblages ' by Martinez and co-authors aims to show the response of the benthic foraminiferal community to by using natural low pH low calcite saturation submarine springs. They show calcareous and agglutinating foraminiferal abundancies decrease, but the calcareous non-symbiont bearing species seem to be impacted the most. The manuscript is reasonably well written and the results are in line with some other similar studies, but I have some concerns about the methodology that could impact the observed trends. Especially lumping porcelaneous and hyaline species together and only using the larger fraction (>250 um) might bias some of the results. The discussion could use some restructuring and extra depth, by for instance analyzing

trends porcelaneous and hyaline species, adding size normalized weights of Discorbis, exploring the effect of salinity and different symbiont species.

Major comments

Page 3 line 14: What kind of substrate was present and was there a difference in substrate near the vents and at the control site? Did you include plants (some benthic species prefer to live on plant debris)?. Did you apply rose bengal staining to only analyze living specimens?

Page 3, line 15: Why did you choose 250 um? Normally 125-150 um is used (Schonfield et al., 2012: Marine Micropaleontology, 94–95), since you might miss the trends in the smaller community now. The trends you observed might be true for larger specimens, but perhaps the smaller specimens tell a different story...

Page 4, line 13-14 I am not sure about 'lumping' low mg forams together with porcelaneous in one group, since it is known from countless studies they respond different to increased pCO2, perhaps due to e.g. solubility of high MgCO3. Did you check if both hyaline and porcelaneous species in this group show similar trends? Otherwise you might be skewing your results, especially since you see no significant change in weight of shells of Discorbis. I would also be very interested to see (relative) abundances of low (e.g. Discorbis), intermediate (Amphistegina, Astergerina) and high Mg species (Quinqueloculina, Archaias) between ojos and control. It would bring something new to the existing studies on different sites, especially since you have the opportunity to test it here on species with very contrasting Mg content.

Discussion section: The authors do not (clearly) explain why the abundancy of agglutinating foraminifera decreases at the vents. They do not calcify or have symbionts, so the explanations given to explain the calcareous response (proton pumping and symbiont activity) do not apply. Could salinity play a role?

Page 7 line 22-29 The authors missed a big overview study by Doo et al., 2014 (Biol.

Bull. 226: 169–186.) in which they present a nice overview of response of larger benthic foraminifera to ocean acidification. I think their discussion would benefit from including these observations. For instance, to look at the different kind of symbionts (diatom, dinos) your foraminifera species have and if they follow the general trend of Doo et al., 2014. It would also be informative to add an overview of the response of benthic foraminifera (symbiont/non symbiont) in different studies, like in Keul et al., 2013 to show how your data fits laboratory and field experiments.

Minor comments

Throughout manuscript pCO2 (p in italics)

Page 2 line 9-10: Keul er al., also contains a nice overview of species-specific responses

Page 2 line 29: Do other chemical parameters change between ojos and control? Oxygen, sulphates?

Page 3, line 08-10: It is more common to use the K1 and K2 values from Lueker et al., 2000. I would suggest recalculating your carbonate parameters with these, since Millero (2010) are known to cause discrepancies in the results amongst programs (for details see Orr et al 2015). Please also specify in more detail what constants were used for carbonate system calculations. For example, what term was used for KHSO4? Dickson (1990) is commonly used.

Page 3, line 14-17: How much gram of sediment was counted?

Page 3, line 20-23: Even though only specimens from 250-355 um were picked, the test weights have to be normalized for size to be able to compare between sites and studies.

Page 4, line 7: There is no seasonality in the output/flux of the vents?

Page 4, line 24-26. The abundance of agglutinating foraminifera is very low already in

the control sites. Do you think the numbers are high enough to make big statements of agglutinating foraminifera being more resilient to low calcite saturation state?

Page 5, line 31: Fig 4 not 5

Page 5, Line 17-20-21: Fig 5 not 4

Discussion: The discussion needs some restructuring, perhaps adding paragraphs might help?

Page 6 line 25: 3-6 units is in my opinion not a 'slight' but a big difference and should be taken into account or at least discussed

Page 8, line 7-10. There is also evidence from culture experiments showing very species specific response of agglutinating foraminfiera with pCO2 (e.g. van Dijk et al., 2017, JFR).

Page 8, line 19-25 This is not really discussed in detail the discussion and has therefore no place in the conclusion. Could you add a paragraph on this in the discussion section.

Table 1: check number of decimals for consistency. Why is there no error on calculated CO2sys values, you could apply a propagating error.

Figure 3: Top three panels: Can you put the 0 on the intersection between y and x axis? Is it possible to order the ojos from e.g. South to North or vice versa?

---

## Short Comment (SC1) · 10 Aug 2018

A. Paytan

apaytan@ucsc.edu

Dear Inge,

Thank you for your thoughtful comments on the paper. Ana is defending her thesis this month but we will be revising the manuscript to address your comments right after the defense. Specifically, we will look again at the smaller size fraction to see if that changes the conclusions. we selected to focus on the large size fractions because there seems to be more material and it was easier to identify the forams.

To answer a few quick questions you had -

Page 3 line 14: What kind of substrate was present and was there a difference in substrate near the vents and at the control site? - substrate is coarse sand at both

locations which are only a few meters apart.

Did you include plants (some benthic species prefer to live on plant debris)?. No we did not include any epiphytes unless they were in the sediment. Because there are no grass beds right at the spring we set the control at a similar site as this made sense for more direct comparison.

Did you apply rose bengal staining to only analyze living specimens? we used rose bengal but pretty much everything got stained so that did not work well. We also tried cell tracker green but in this case only very few forams got stained so that was not useful for the statistics.

Page 3, line 15: Why did you choose 250 um? Normally 125-150 um is used (Schonfield et al., 2012: Marine Micropaleontology, 94–95), since you might miss the trends in the smaller community now. The trends you observed might be true for larger specimens, but perhaps the smaller specimens tell a different story:

Note that the title is reflective of the fact that we looked at the large fraction but we will check the rest for comparison.

Page 4, line 13-14 I am not sure about 'lumping' low mg forams together with porcelaneous in one group, since it is known from countless studies they respond different to increased pCO2, perhaps due to e.g. solubility of high MgCO3. Did you check if both hyaline and porcelaneous species in this group show similar trends? Otherwise you might be skewing your results, especially since you see no significant change in weight of shells of Discorbis. I would also be very interested to see (relative) abundances of low (e.g. Discorbis), intermediate (Amphistegina, Astergerina) and high Mg species (Quinqueloculina, Archaias) between ojos and control. It would bring something new to the existing studies on different sites, especially since you have the opportunity to test it here on species with very contrasting Mg content.

We have the information by species and will check to see abundance vs Mg/Ca content.

Discussion section: The authors do not (clearly) explain why the abundancy of agglutinating foraminifera decreases at the vents. They do not calcify or have symbionts, so the explanations given to explain the calcareous response (proton pumping and symbiont activity) do not apply. Could salinity play a role?

The salinity difference is small but and this does not seem to be the case in the correlations trends maybe it is related to higher energy environment close to the ojos..... However as you said the abundance of agglutinating foraminifera is very low already in the control sites so most likely the numbers are not high enough to make big statements of agglutinating foraminifera being more resilient to low calcite saturation state. and we will qualify this statement.

Page 7 line 22-29 The authors missed a big overview study by Doo et al., 2014 (Biol.Bull. 226: 169–186.) in which they present a nice overview of response of larger benthic foraminifera to ocean acidification. I think their discussion would benefit from including these observations. For instance, to look at the different kind of symbionts (diatom, dinos) your foraminifera species have and if they follow the general trend of Doo et al., 2014. It would also be informative to add an overview of the response of benthic foraminifera (symbiont/non symbiont) in different studies, like in Keul et al., 2013 to show how your data fits laboratory and field experiments.

Thanks good suggestions to look into

Minor comments will be corrected

---

## Referee Comment (RC2) · Anonymous Referee #2 · 26 Aug 2018

The paper by Martinez et al. describes a very interesting study in which natural variability in carbonate saturation state at submarine springs (ojos) is used to assess the benthic foraminiferal response to ocean acidification. The authors find that proximity to submarine springs impacts the benthic foraminiferal community. In particular, they find a decrease in overall abundances, but also that symbiotic calcareous species are less affected than non-symbiont bearing species, and agglutinated foraminifera may be least impacted. The paper is overall well written, and the conclusions are interesting. However, in some areas, the complexities and richness of the underlying dataset could be better served. Digging further into some of the complexities here should allow the authors to better support their current conclusions, but I suspect it will also lead to some more specific and novel results. Overall, there are three major (somewhat

related) issues, which I would strongly urge the authors to address.

1) First, is the assumption that proximity to "ojos" impacts benthic foraminifera entirely or primarily due to difference in calcite saturation state from the ambient environment. This does not seem like a foregone conclusion to me. These are essentially isolated regions of increased fresh-water influence in a marine context and could be different in a number of ways from the surrounding environment. The authors mention, but then rapidly dismiss, the salinity differences between the ojos and ambient environment (6:24-28). However to entirely dismiss salinity requires either a more detailed quantitative analysis to try to tease apart these covarying parameters, and/or a more in depth discussion of the known sensitivities of different foraminifera and communities to salinity. There could be several additional environmental differences, such as oxygenation, or changes in nutrient or metal concentrations from terrestrial sources that could produce sensitivities in some species (I might look into the literature on benthic foraminifera communities as tracers of metal contamination). Finally, there could be differences in benthic community or environment (substrate? Food source? predation?) between the ojos and control sites. All of this should be discussed. 2) A broad view of how major groups of foraminifera respond within the community (symbiotic, agglutinated, etc.) is much needed and well served by the study design. However, it is a shame that it comes at the expense of a discussion of a species, clade, or more finely-defined functional group level response. This study would be more impactful if it also reported the species-level assemblages at each sites. Are there specific species or genera that appear more or less robust to the environmental differences between ojo and "control" environments? Such a discussion is especially warranted given the species-level differences in response to ocean acidification that have repeatedly been shown in culture studies. 3) Finally, there appear to be clear differences between ojo/control pairings which are occasionally mentioned in passing, but never fully addressed. For example, looking at Figure 2, I am immediately greeted with some pretty basic questions such as "Why is there such a large difference in abundance at Mini and its control compared to Gorgos and its control?" and "What is different about Norte that

the low-saturation abundance is as high as the high-saturation groups at other sites?" If saturation state is the primary driver of total abundance this should be an unexpected result! Without further information or context about either the assemblages or environments at each site, it is hard to even start thinking about some of these complexities. There is a lot to uncover here that may still require some further analyses.

Minor points: - Why the use of the >255 size fraction? Could this have biased the results especially is different size species respond differently? For example the Henehan et al., 2017 paper on weight suggests size may impact species calcification response. Is it possible that smaller species may have differing metabolic requirements? - What was the depth of each site? I almost wonder if this could be contributing to some of the inter-site differences? - Can you report all species identified (in addition to the most abundant)? It would be very valuable for assessing assemblages at a finer scale and also for future workers. Ideally, it would be good to see full assemblages reported at each site and represented and compared in a figure. - How large were the sediment samples from which forams were measured? Did it differ between sites? And how were they collected? Importantly, how deep into the sediment were samples taken and was consistency in this regard maintained across sites? - Page 2, Line 12-3: It is also worth noting that some species also tolerate (even specialize in) high CO2 environments such as oxygen minimum zones – look into some of the Bernhard papers on this in Santa Barbara Basin– and low salinity (low saturation state) estuaries. - Page 3, Line 12: What is your balance error? - Sections 3.3. and 3.4 raise a lot of questions for the reader about what is producing the reported differences between sites. See major point 3, but I think the conclusions could be made sounder if some of these type of questions are tackled. - Section 3.5: Again, it looks as if there may be a difference between sites. Is this statistically significant? Also, this should refer to Figure 4. - Page 5L Line 1: "but not Gorgos" - Page 6, Lines 23-34: "Therefore, while abundant CT may help lower the potential impact on foraminiferal calcification at low pH, it does not seem to fully counteract the effect of low $\Omega$ ." I don't think this is quite right. If I understand correctly, the authors are arguing that this important parameter is carbonate ion concentration/saturation state, and total inorganic carbon and pH are important drivers of this both intra- and extra-cellularly. If so, this should read along the lines of "Therefore, while abundant CT may increase the availability of carbonate ion, it does not seem to fully counteract the effect of low pH on $\Omega$." - Page 7, lines 1-3: This really glosses over the huge history and literature on shell weight and carbonate chemistry in planktonic foraminifera. Have a look at Table 7 in Weinkauf et al., 2016 for a good (though not exhaustive) review of some of this work. - Page 7, lines 8-10: Davis et al., 2017 also shows variable individual responses to saturation state within a population of foraminifera. - Many of the figures appear low quality and pixelated. This may be the result of embedding, but double check. Also, why not include color as well as gray-scale for this online pupblication? - For figures 2-4, it would be useful to have represented on these plots which groups are significantly different from one another (as in the Results section). Consider including this?

---

## Short Comment (SC2) · 26 Aug 2018

Thanks for the great input and suggestions. Ana is defending her thesis this week and we will get back to the samples, data and figures to address your comments and suggestions as soon as she is defends hopefully successfully.

Adina

---

## Referee Comment (RC3) · Anonymous Referee #3 · 11 Sep 2018

Review of "Impact of carbonate saturation on large Caribbean benthic foraminifera assemblages" by Martinez et al. for Biogeosciences

In order to assess the impact of carbonate saturation on the assemblages of large benthic foraminifera in the Caribbean, Martinez et al. compare assemblages at low pH, low calcite saturation submarine spring sites with control sites of higher calcite saturation. This is an important question to tackle given that carbonate saturation will likely decrease in the future due to the increased impacts of ocean acidification. This is a unique experimental setup to take advantage of a natural location where these impacts can be studied. The authors find that at the low pH sites, there is a decrease in total benthic abundance, and increase in symbiont bearing species, and an increase in agglutinated species. Overall, non-symbiont bearing species may be more sensitive

to the impacts of ocean acidification. The paper is well written and organized well, and I have only a couple of comments that I believe the authors can easily address. I should note that I am not a specialist in large benthic foraminifera, so I hope that the other reviewers may have more specific points about that.

Key points: (1) One of my main concerns with the study is that the authors are quick to dismiss that there may be other environmental differences between the submarine springs and the control sites, and perhaps too simplistically conclude that the carbonate saturation (and pH) differences are the main control on the foraminifera assemblage differences. For example, there are large salinity differences between the sites that I think warrants more discussion. Are there any differences in food sources, turbidity, depths, etc? (2) The authors choose to analyze the >250 micrometer fraction of sediment, but do not explain their choice for this. I think that by choosing this fraction, they may be omitting smaller, important foraminifera from their analyses. One of the potential impacts of decreased carbonate saturation is that foraminifera may be smaller. So, it may be that by looking at this larger size fraction, they are missing foraminifera that may be smaller at the submarine spring sites but may still be present. It would be very helpful is the authors can repeat some analyses using a >150 micrometer fraction, for example.

---

## Short Comment (SC3) · 11 Sep 2018

We thank the reviewer for the useful comments and will respond to them along with the other comments from the other two reviewers. Specifically, we are looking into the smaller size fraction to see if we can obtain new information. Please do note that the title specifies LARGE foraminifera

Thanks Adina

---

## Author Comment (AC1) · 22 Oct 2018

Dear Inge,

Please find our response to your very helpful comments which improved the manuscript considerbly. Each response in listed directly following the comments made

Referee #1 General Comments:

Comment 1: The manuscript 'Impact of carbonate saturation on large Caribbean benthic foraminifera assemblages' by Martinez and co-authors aims to show the response of the benthic foraminiferal community to by using natural low pH low calcite saturation submarine springs. They show calcareous and agglutinating foraminiferal abundancies decrease, but the calcareous non-symbiont bearing species seem to be impacted the

most. The manuscript is reasonably well written, and the results are in line with some other similar studies, but I have some concerns about the methodology that could impact the observed trends. Especially lumping porcelaneous and hyaline species together and only using the larger fraction (>250 um) might bias some of the results. The discussion could use some restructuring and extra depth, by for instance analyzing trends porcelaneous and hyaline species, adding size normalized weights of Discorbis, exploring the effect of salinity and different symbiont species.

Reply: We thanks Dr. van Dijk for recognizing the importance of the study and we appreciate the suggestions to include a more detailed analysis of the foraminifera data. We have included new analyses and interpretation of abundance of porcelaneous and hyaline foraminifera and of symbiotic diatom-bearing and chlorophyte-bearing foraminifera in the new version of the manuscript. We have also included an explanation of why the >250 ïA■m size fraction was used in the analysis and a deeper discussion of the effects of salinity on foraminifera. Although we do not report size normalized weights we did make an effort to select individuals that were similar in size as much as possible and re re-weighted many samples to see if this changes the results which it did not. Regardless we acknowledge this shortcoming of not normalizing the weights. The updated discussion was structured in paragraphs to facilitate readability.

Major comments:

Comment 2. Page 3 line 14: What kind of substrate was present and was there a difference in substrate near the vents and at the control site? Reply: The substrate is coarse sand at all locations, and control and ojo sites were only a few meters apart. We have included this information on the methods section. Did you include plants (some benthic species prefer to live on plant debris)?

Reply: We did not sample plants specifically and any epiphytes that are included were in the upper sediment. Because there are no grass beds right at the springs, we set the control sites to be as similar as possible to the ojo sites avoiding grass beds as this

made sense for more direct comparison.

Did you apply rose bengal staining to only analyze living specimens?

Reply: We used Rose Bengal but pretty much everything got stained to some degree and it was hard to distinguish dead from live using this stain. Rose Bengal can stain proteins of dead specimens that are not fully decomposed, or proteins of bacteria inside or on the tests, producing false positives that overestimate abundance of foraminifera (Bernhard et al., 2006; Paleoceanography, vol. 21, pa4210, doi:10.1029/2006PA001290, 2006). In addition, it is hard to distinguish the stained specimens in some species with opaque tests such as Archaias angulatus (Wantland, 1967). We also tried CellTracker Green but, in this case, only very few forams got stained so that was not useful for the statistical analysis.

Comment 3. Page 3, line 15: Why did you choose 250 um? Normally 125-150 um is used (Schonfield et al., 2012: Marine Micropaleontology, 94–95), since you might miss the trends in the smaller community now. The trends you observed might be true for larger specimens, but perhaps the smaller specimens tell a different story. . .

Reply: We focus on the large size fraction and clearly note this in the title because this size fraction constituted the majority of foraminifera in the samples. Indeed, many foraminifera typically found in tropical lagoons attain large sizes and have mortality rates of above 95% of juveniles until they reach a diameter of 0.5 mm (Why are larger forams large? Hallock, 1985, Paleobiology) which may explain the low abundance of the smaller sized forams in our samples. We now described in the methods section that analyses of the <250 um fraction we found only 9-27 specimens per gram sediment while in the >250 um fraction around 300-500 specimens were found. The fraction of >250 represents the adult individuals more prone to be preserved in the sediment (Martin, 1986). We have included an explanation of the size fraction selection in the methods section.

Comment 4. Page 4, line 13-14 I am not sure about 'lumping' low Mg forams together

with porcelaneous in one group, since it is known from countless studies they respond different to increased pCO2, perhaps due to e.g. solubility of high MgCO3. Did you check if both hyaline and porcelaneous species in this group show similar trends? Otherwise you might be skewing your results, especially since you see no significant change in weight of shells of Discorbis. I would also be very interested to see (relative) abundances of low (e.g. Discorbis), intermediate (Amphistegina, Astergerina) and high Mg species (Quinqeloculina, Archaias) between ojos and control. It would bring something new to the existing studies on different sites, especially since you have the opportunity to test it here on species with very contrasting Mg content.

Reply: We have included the absolute and relative abundance of porcelaneous and hyaline foraminifera as well as of low, intermediate and high magnesium foraminifera in the new version of the manuscript.

Comment 5. Discussion section: The authors do not (clearly) explain why the abundancy of agglutinating foraminifera decreases at the vents. They do not calcify or have symbionts, so the explanations given to explain the calcareous response (proton pumping and symbiont activity) do not apply. Could salinity play a role?

Reply: The absolute abundance of agglutinating foraminifera did not differ with saturation state in 3 of the 5 sampled submarine springs and we note that in the paper. The relative abundance was higher at low saturation than at high saturation at one site and did not differ in the other 4 sampled sites. We have rewritten these results in a clearer way and we have discussed why agglutinated foraminifera are not sensitive to carbonate saturation as the reviewer indicates. We also explained why we think salinity is not driving changes in abundance of any of the foraminifera at the springs. In addition, sensors deployed at the springs showed that salinity is >30psu over 90% of the time and it does not drop below 27psu at the sites we sampled. When salinity drops below 30 psu (7% of the time), the low salinity exposure lasts for very short periods of time always less than 1 hour (Crook et al., Supporting Information, PNAS July 2, 2013 110 (27) 11044-11049; https://doi.org/10.1073/pnas.1301589110). Based on literature the

majority of forams we found have very wide salinity tolerance as they are common in settings that have variable salinity such as close to shore and in lagoons.

Comment 6. Page 7 line 22-29 The authors missed a big overview study by Doo et al., 2014 (Biol. Bull. 226: 169–186.) in which they present a nice overview of response of larger benthic foraminifera to ocean acidification. I think their discussion would benefit from including these observations. For instance, to look at the different kind of symbionts (diatom, dinos) your foraminifera species have and if they follow the general trend of Doo et al., 2014. It would also be informative to add an overview of the response of benthic foraminifera (symbiont/non symbiont) in different studies, like in Keul et al., 2013 to show how your data fits laboratory and field experiments.

Reply: We have now included the absolute and relative abundance of diatom-bearing (Ampistegina and Asterigerina) and chlorophyte-bearing foraminifera (Archaias). The studies included in the review by Doo et al., 2014 did not study chlorophyte-bearing foraminifera, therefore we cannot compare our results to the trends seen in other studies that only focused on diatom and dinoflagellate bearing foraminifera. We have included discussion on the potential effects of symbionts on foraminifera calcification.

Minor comments

Comment 7. Throughout manuscript pCO2 (p in italics)

Reply: We changed p to italics throughout the text.

Comment 8. Page 2 line 9-10: Keul er al., also contains a nice overview of species-specific responses

Reply: We have added this relevant reference in the introduction.

Comment 9. Page 2 line 29: Do other chemical parameters change between ojos and control? Oxygen, sulphates?

Reply: There are some relatively small differences between ojos, for example the water

discharging at ojo Norte has lower oxygen and it is slightly more reducing than the other ojos during very low tide conditions. However, we do not have replicates of ojos that differ from each other (in fact Norte is the only that is slightly different than the other ojos) hence we cannot do a comprehensive analysis on the impact of these differences in chemistry. Specifically, we did not see any unique trends at ojo Norte hence we do not attribute this to the small difference in water chemistry. Regardless we emphasize throughout that there are advantages and disadvantages to conducting field observation with the main issue if that there are confounding variables but on the other hand the results we obtain cannot be replicated in laboratory settings and are more realistic.

Comment 10. Page 3, line 08-10: It is more common to use the K1 and K2 values from Lueker et al., 2000. I would suggest recalculating your carbonate parameters with these, since Millero (2010) are known to cause discrepancies in the results amongst programs (for details see Orr et al 2015). Please also specify in more detail what constants were used for carbonate system calculations. For example, what term was used for KHSO4? Dickson (1990) is commonly used.

Reply: Thanks for this important suggestion. We recalculated the carbonate chemistry parameters with K1 and K2 from Lueker et al. 2000 and included a more detailed description of the constants used (KHSO4 from Dickson 1990 and total boron from Uppström, 1974) in the methods section.

Comment 11. Page 3, line 14-17: How much gram of sediment was counted?

Reply: At least 1gram of sediment and on average 2 grams of sediment (per replicate) was analyzed. We have inserted this information in the methods section.

Comment 12. Page 3, line 20-23: Even though only specimens from 250-355 um were picked, the test weights have to be normalized for size to be able to compare between sites and studies.

[Figure]

Reply: We agree with the reviewer that this would be useful. However, we have not done this and in an attempt to resolve this issue we re-analyzed 7 representative samples of the 50 sediment samples we collected for this study (5 replicates at 5 ojos and 5 control sites) normalizing to size, and still did not find any statistically significant difference in the weight. It seemed to be a major waste of time and effort to re-analyze again all 50 samples. We report on that and acknowledge the need to do so in the manuscript.

Comment 13. Page 4, line 7: There is no seasonality in the output/flux of the vents?

Reply: Yes, there is an increase in groundwater discharge during the rainy season and during low tide. We refer to a paper that descries the variability at the site. We note however that the foraminifera in the upper sediments represent decades or longer and these organisms grew under all the different conditions at the sites.

Comment 14. Page 4, line 24-26. The abundance of agglutinating foraminifera is very low already in the control sites. Do you think the numbers are high enough to make big statements of agglutinating foraminifera being more resilient to low calcite saturation state?

Reply: It is true that the abundance of agglutinating foraminifera is very low already in the control sites and most likely the numbers are not high enough to make big statements of agglutinating foraminifera; however, our data is in agreement with other high pCO2 field studies in Papua New Guinea (Uthicke et al., 2013) and Italy (Dias, 2010). We note that they are rare but still report on the results as we think this is useful.

Comment 15. Page 5, line 31: Fig 4 not 5

Reply: Thank you for spotting this mistake, we changed it in the newest version of the manuscript.

Comment 16. Page 5, Line 17-20-21: Fig 5 not 4

Reply: We changed this mistake in the newest version of the manuscript.

Comment 17. Discussion: The discussion needs some restructuring, perhaps adding paragraphs might help?

Reply: As the reviewer suggested the discussion was divided into different sections to improve the organization of information.

Comment 18. Page 6 line 25: 3-6 units is in my opinion not a 'slight' but a big difference and should be taken into account or at least discussed

Reply: We have included further discussion on salinity effects to make it clearer in the manuscript (see reply to comment 5).

Comment 19. Page 8, line 7-10. There is also evidence from culture experiments showing very species specific response of agglutinating foraminfiera with pCO2 (e.g. van Dijk et al., 2017, JFR).

Reply: Thank you for the suggestion. This is a relevant paper that has been included in the discussion of impacts of carbonate saturation on agglutinating foraminifera.

Comment 20. Page 8, line 19-25 This is not really discussed in detail the discussion and has therefore no place in the conclusion. Could you add a paragraph on this in the discussion section.

Reply: We moved and rewrote these lines in the discussion section.

Comment 21. Table 1: check number of decimals for consistency. Why is there no error on calculated CO2sys values, you could apply a propagating error.

Reply: We updated the number of decimals for consistency. We included the std on the actual reported values that we measured in the field using these values if anyone is interested the error of the calculated values (pH and carbonate saturation) could be determined. However, we did not include this because the difference between the ojos and control are so large that this will not really change any of the conclusions and discussion.

Comment 22. Figure 3: Top three panels: Can you put the 0 on the intersection between y and x axis?

Reply: The new plots have now the same Y-axis scale and are all aligned at 0.

Is it possible to order the ojos from e.g. South to North or vice versa?

Reply: The sites are now organized from North (Norte) to South (Gorgos) in plots and tables.

Thank you Adina
* * *

---

## Author Comment (AC2) · 22 Oct 2018

Dear Anonymous Referee #2

Thank you for your very insightful comments which improved the paper. Below we address each comment (following each comment).

The paper by Martinez et al. describes a very interesting study in which natural variability in carbonate saturation state at submarine springs (ojos) is used to assess the benthic foraminiferal response to ocean acidification. The authors find that proximity to submarine springs impacts the benthic foraminiferal community. In particular, they find a decrease in overall abundances, but also that symbiotic calcareous species are less affected than non-symbiont bearing species, and agglutinated foraminifera may

[Figure]

be least impacted. The paper is overall well written, and the conclusions are interesting. However, in some areas, the complexities and richness of the underlying dataset could be better served. Digging further into some of the complexities here should allow the authors to better support their current conclusions, but I suspect it will also lead to some more specific and novel results. Overall, there are three major (somewhat related) issues, which I would strongly urge the authors to address.

Reply: We thank the reviewer for noting that the paper is interesting and original and suggesting areas for improvement which we have addressed in the revised version.

Comment 1) First, is the assumption that proximity to "ojos" impacts benthic foraminifera entirely or primarily due to difference in calcite saturation state from the ambient environment. This does not seem like a foregone conclusion to me. These are essentially isolated regions of increased fresh-water influence in a marine context and could be different in a number of ways from the surrounding environment. The authors mention, but then rapidly dismiss, the salinity differences between the ojos and ambient environment (6:24-28). However to entirely dismiss salinity requires either a more detailed quantitative analysis to try to tease apart these covarying parameters, and/or a more in depth discussion of the known sensitivies of different foraminifera and communities to salinity. There could be several additional environmental differences, such as oxygenation, or changes in nutrient or metal concentrations from terrestrial sources that could produce sensitivities in some species (I might look into the literature on benthic foraminifera communities as tracers of metal contamination). Finally, there could be differences in benthic community or environment (substrate? Food source? predation?) between the ojos and control sites. All of this should be discussed.

Reply: Sensors deployed at these specific ojos determined that salinity is above 30 for 93% of the time, and when it drops below 30, it is for short periods of time of less than 1 hour and does not fall below 27 (Crook et al., Supporting Information, PNAS July 2, 2013 110 (27) 11044-11049; https://doi.org/10.1073/pnas.1301589110). We emphasize this and refer to the above study. In addition, we note that many of the

species in our study have a very broad salinity tolerance range as typical to shallow coastal lagoon and estuary settings. We have included the salinity sensitivities of the major species in our study for which data is available. We also compared our results to results from laboratory experiments where only carbon chemistry is changed to support our idea that the carbonate saturation is the main driver of foraminiferal abundances we see in our field site. Regarding other variables, the ojo and control sites are just a few meters apart with identical substrate (coarse sand), water depth and light, and while we did not monitor at such close proximity we expect that predation and food sources are also similar. Regarding other chemical differences in the discharging water indeed there are small differences in nutrients and oxygen but not in trace metals. Specifically, ojo Norte has lower oxygen during very low tides and slightly more reducing conditions but there was nothing particularly unique about this ojo in terms of the trends observed. Moreover, the differences among ojos were smaller than the ranges of variability within each ojo (see also reply to reviewer #1). Finally, each ojo is slightly different but we do not have replicates of identical ojos so we cannot deduce statistical differences and attribute them to such conditions. As we write this is the nature of doing field work there are confounding variables, but you get more realistic results.

Comment 2) A broad view of how major groups of foraminifera respond within the community (symbiotic, agglutinated, etc.) is much needed and well served by the study design. However, it is a shame that it comes at the expense of a discussion of a species, clade, or more finely-defined functional group level response. This study would be more impactful if it also reported the species-level assemblages at each sites. Are there specific species or genera that appear more or less robust to the environmental differences between ojo and "control" environments? Such a discussion is especially warranted given the species-level differences in response to ocean acidification that have repeatedly been shown in culture studies.

Reply: We have included the contribution and trends in relative abundance of each genus, as well as abundances depending on test type (porcelaneous, hyaline, agglutinated), magnesium content (low, intermediate and high), feeding type (symbiont-bearing, symbiont-barren) and symbiont type (diatom and chlorophyte).

Comment 3) Finally, there appear to be clear differences between ojo/control pairings which are occasionally mentioned in passing, but never fully addressed. For example, looking at Figure 2, I am immediately greeted with some pretty basic questions such as "Why is there such a large difference in abundance at Mini and its control compared to Gorgos and its control?" and "What is different about Norte that the low-saturation abundance is as high as the high-saturation groups at other sites?" If saturation state is the primary driver of total abundance this should be an unexpected result! Without further information or context about either the assemblages or environments at each site, it is hard to even start thinking about some of these complexities. There is a lot to uncover here that may still require some further analyses.

Reply: As noted above, each ojo is slightly different than the other in terms of water chemistry and discharge rates but we do not have replicates of identical ojos so we cannot deduce statistical differences and attribute them to such conditions. As we note in our response to reviewer #1 and in the paper, this is the nature of doing field work there are confounding variables which complicate interpretation, but the results you get from such studies are more realistic. We note that despite the differences between ojos there are common trends and we think it is more useful to focus on these observations than to over analyze differences which would be speculative at best to explain. Complexities are the nature of such studies yet we can still glean useful information.

Minor points:

Comment 4. Why the use of the >255 size fraction? Could this have biased the results especially is different size species respond differently? For example the Henehan et al., 2017 paper on weight suggests size may impact species calcification response. Is it possible that smaller species may have differing metabolic requirements?

[Figure]

Reply: Indeed size is important and can impact metabolism and calcification response however, tropical benthic foraminifera are characterized by large sizes (Why are larger forams large? Hallock, 1985, Paleobiology) and the size fraction >250 um represents the adult individuals likely to be preserved in the sediment (Martin, 1986) since juvenile mortality rates are higher than 95% (Hallock, 1985). We have inserted this information in the methods section. Specifically, in our samples in the > 250 um size fraction we around 300-500 specimens per gram sediment while only 9-27 specimens were found on the fraction of 125-250um hence the smaller size fraction included at most 10% of the foraminifera. Regardless although the larger size fraction represents ∼90% of the forams we specifically refer to Large Foraminifera in the title to be honest to our data. Nonetheless, we have included in the discussion a section on the role of a larger size on increased symbiont concentration and dissolution resistance in sediments (Hönisch et al, 2004), which may be responsible for the changes in abundance we see.

Comment 5. What was the depth of each site? I almost wonder if this could be contributing to some of the inter-site differences?

Reply: The depth ranged from ∼5 to 7 m at all sites. The depth of each site has been included in the water chemistry table. The setting is quite similar, and all sites have similar light conditions.

Comment 6. Can you report all species identified (in addition to the most abundant)? It would be very valuable for assessing assemblages at a finer scale and also for future workers. Ideally, it would be good to see full assemblages reported at each site and represented and compared in a figure.

Reply: We did exploratory analyses in 10% of the samples to determine what the most abundant genera were, and we have now mentioned other genera present in the samples in the results section (Borelis, Clavulina, Elphidium, Spiroloculina, Peneroplis, Laevipeneroplis, Planorbulina, Sorites, Vertebralina and Heterostegina). However, we did not analyze the full assemblage of all the species present in all the samples. There

is already existing literature on full assemblages in the area which found similar results (Wantland, 1967; Triffleman et al, 1991; Gischler et al., 2008).

Comment 7. How large were the sediment samples from which forams were measured? Did it differ between sites? And how were they collected? Importantly, how deep into the sediment were samples taken and was consistency in this regard maintained across sites?

Reply: The samples were collected from the upper centimeter of sediment with a spoon and a centrifuge tube across all sampled sites. We analyzed at least 1 gram of sediment per replicate, with an average of ∼2 grams of sediment per replicate. We aimed for at least 300 individuals per sample; however, due to the low abundances in some of the samples (especially in samples collected at springs), this was not always possible and 24 of the 50 samples had less than 300 individuals per gram. We have included this information in the methods section.

Comment 8. Page 2, Line 12-3: It is also worth noting that some species also tolerate (even specialize in) high CO2 environments such as oxygen minimum zones – look into some of the Bernhard papers on this in Santa Barbara Basin– and low salinity (low saturation state) estuaries.

Reply: We note in the introduction that foraminifera are versatile and that some tolerate high CO2 settings and low salinity. Note however that most OMZs and also Santa Barbara basin are above the CCD and are super saturated with respect to calcite. Most studies of benthic forams in OMZs and also in SBB focus on the low oxygen rather than high CO2.

Comment 9. Page 3, Line 12: What is your balance error?

Reply: The analytical micro-balance has an error of $\pm$ 5 ïĄ■g. This information has been inserted in the methods section.

Comment 10. Sections 3.3. and 3.4 raise a lot of questions for the reader about what

is producing the reported differences between sites. See major point 3, but I think the conclusions could be made sounder if some of these type of questions are tackled.

Reply: See reply to major point 3. We agree that it would be nice to address this but we do not think it is possible since the slight differences in other water chemistry parameters are not consistent between ojos so no statistical power to decipher the causes for the difference in magnitude of the responses.

Comment 11. Section 3.5: Again, it looks as if there may be a difference between sites. Is this statistically significant? Also, this should refer to Figure 4.

Reply: Thanks for finding the mistake, we have added new plots and changed the number of the figures.

Comment 12. Page 5L Line 1: "but not Gorgos" - Page 6, Lines 23-34: "Therefore, while abundant CT may help lower the potential impact on foraminiferal calcification at low pH, it does not seem to fully counteract the effect of low ." I don't think this is quite right. If I understand correctly, the authors are arguing that this important parameter is carbon- ate ion concentration/saturation state, and total inorganic carbon and pH are important drivers of this both intra- and extra-cellularly. If so, this should read along the lines of "Therefore, while abundant CT may increase the availability of carbonate ion, it does not seem to fully counteract the effect of low pH on ."

Reply: We removed this paragraph from the paper however elsewhere in the paper when we comment about the CT we changed the sentence as suggested.

Comment 13. Page 7, lines 1-3: This really glosses over the huge history and literature on shell weight and carbonate chemistry in planktonic foraminifera. Have a look at Table 7 in Weinkauf et al., 2016 for a good (though not exhaustive) review of some of this work.

Reply: Thanks for the reference to this interesting paper, we have rewritten the test weight discussion to be more succinct and straightforward.

[Figure]

Comment 14. Page 7, lines 8-10: Davis et al., 2017 also shows variable individual responses to saturation state within a population of foraminifera.

Reply: Thank you for this relevant reference, we have inserted this reference in the discussion of test weights.

Comment 15- Many of the figures appear low quality and pixelated. This may be the result of embedding, but double check. Also, why not include color as well as gray-scale for this online publication?

Reply: We have changed the figures to color bar plots and increased image quality to improve readability.

Comment 16. For figures 2-4, it would be useful to have represented on these plots which groups are significantly different from one another (as in the Results section). Consider including this?

Reply: We have added asterisk to significant differences ($p < 0.05$) between paired control and spring sites.

Thank you for the time you devoted to improve the paper

Adina

---

## Author Comment (AC3) · 22 Oct 2018

Dear Referee #3,

Thank you for your useful input. Below we copy your comments and our response to them follows

In order to assess the impact of carbonate saturation on the assemblages of large benthic foraminifera in the Caribbean, Martinez et al. compare assemblages at low pH, low calcite saturation submarine spring sites with control sites of higher calcite saturation. This is an important question to tackle given that carbonate saturation will likely decrease in the future due to the increased impacts of ocean acidification. This is a unique experimental setup to take advantage of a natural location where these

impacts can be studied. The authors find that at the low pH sites, there is a decrease in total benthic abundance, and increase in symbiont bearing species, and an increase in agglutinated species. Overall, non-symbiont bearing species may be more sensitive to the impacts of ocean acidification. The paper is well written and organized well, and I have only a couple of comments that I believe the authors can easily address.

We thank the reviewer for this nice summary of our paper she/he brings up similar concerns are the two other reviewers and we have addressed these issues in the new version.

Key points: (1) One of my main concerns with the study is that the authors are quick to dismiss that there may be other environmental differences between the submarine springs and the control sites, and perhaps too simplistically conclude that the carbonate saturation (and pH) differences are the main control on the foraminifera assemblage differences. For example, there are large salinity differences between the sites that I think warrants more discussion. Are there any differences in food sources, turbidity, depths, etc?

Reply: We have expanded the discussion on the confounding variables when working in natural settings. Specifically, we note that in selecting sites we tried as much as possible that the ojos and control sites will be as similar as possible in all other aspects (depth, substrate, light, currents, temperatures etc.) but the water carbonate chemistry. Salinity to some degree co-varies with the carbonate parameters but the difference in salinity is relatively small between the ojo water and the controls at the sites we selected. As noted in the response to the other 2 reviewers we expanded the explanation about salinity, mentioned the advantages and limitations of field work and noted that by comparing our results to those obtained in controlled laboratory experiments and at other locations we gain confidence in our conclusions.

(2) The authors choose to analyze the >250 micrometer fraction of sediment, but do not explain their choice for this. I think that by choosing this fraction, they may be omitting

smaller, important foraminifera from their analyses. One of the potential impacts of decreased carbonate saturation is that foraminifera may be smaller. So, it may be that by looking at this larger size fraction, they are missing foraminifera that may be smaller at the submarine spring sites but may still be present. It would be very helpful is the authors can repeat some analyses using a >150 micrometer fraction, for example.

Reply: We chose the >250 size fraction because it represents the adult foraminifera assemblage likely to be preserved in the sediment (Martin, 1986) since tropical benthic foraminifera are characterized by large sizes (Hallock, 1985). This fraction comprises 90% of the forams in our samples, probably due to the high mortality rates of juveniles (>95%, Hallock, 1985). We have now inserted this information in the text. See detailed response to reviewers 1 and 2.

Thanks for contributing to improve the manuscript

Adina